# ATP-citrate lyase promotes axonal transport across species

Aviel Even [1,10], Giovanni Morelli[2,3,10], Silvia Turchetto[2,10], Michal Shilian[1], Romain Le Bail[2], Sophie Laguesse[2], Nathalie Krusy[2], Ariel Brisker[1], Alexander Brandis[4], Shani Inbar[1], Alain Chariot[5], Frédéric Saudou [6,7,8], Paula Dietrich[9], Ioannis Dragatsis[9], Bert Brone [3], Loïc Broix [2], Jean-Michel Rigo [3], Miguel Weil[1,11✉] & Laurent Nguyen [2,11✉]

Microtubule (MT)-based transport is an evolutionary conserved process finely tuned by posttranslational modifications. Among them, α-tubulin acetylation, primarily catalyzed by a vesicular pool of α-tubulin N-acetyltransferase 1 (Atat1), promotes the recruitment and processivity of molecular motors along MT tracks. However, the mechanism that controls Atat1 activity remains poorly understood. Here, we show that ATP-citrate lyase (Acly) is enriched in vesicles and provide Acetyl-Coenzyme-A (Acetyl-CoA) to Atat1. In addition, we showed that Acly expression is reduced upon loss of Elongator activity, further connecting Elongator to Atat1 in a pathway regulating α-tubulin acetylation and MT-dependent transport in projection neurons, across species. Remarkably, comparable defects occur in fibroblasts from Familial Dysautonomia (FD) patients bearing an autosomal recessive mutation in the gene coding for the Elongator subunit ELP1. Our data may thus shine light on the pathophysiological mechanisms underlying FD.

[1] Laboratory for Neurodegenerative Diseases and Personalized Medicine, The Shmunis School of Biomedicine and Cancer Research, The George S. Wise Faculty for Life Sciences, Sagol School of Neurosciences, Tel Aviv University, Ramat Aviv 69978, Israel. [2] Laboratory of Molecular Regulation of Neurogenesis, GIGA-Stem Cells, Interdisciplinary Cluster for Applied Genoproteomics (GIGA-R), University of Liège, C.H.U. Sart Tilman, Liège 4000, Belgium. [3] BIOMED Research Institute, Hasselt 3500, Belgium. [4] Life Sciences Core Facilities, Weizmann Institute of Science, Rehovot, Israel. [5] Laboratory of Medical Chemistry, GIGA-Stem Cells, Interdisciplinary Cluster for Applied Genoproteomics (GIGA-R), University of Liège, C.H.U. Sart Tilman, Liège 4000, Belgium. [6] Univ. Grenoble Alpes, Inserm, U1216, CHU Grenoble Alpes, Grenoble Institut Neurosciences, 38000 Grenoble, France. [7] Inserm, U1216, F-38000 Grenoble, France. [8] CHU Grenoble Alpes, F-38000 Grenoble, France. [9] Department of Physiology, University of Tennessee Health Science Center, Memphis, TN 38163, USA. [10] These authors contributed equally: Aviel Even, Giovanni Morelli, Silvia Turchetto. [11] These authors jointly supervised this work: Miguel Weil, Laurent Nguyen. ✉email: miguelw@tauex.tau.ac.il; lnguyen@uliege.be

Axonal transport is an evolutionary conserved process that delivers cargoes to distant subcellular compartments. It is supported by molecular motors (kinesins and dyneins) running along microtubule (MT) tracks and is particularly important for projection neurons that send axons to distant targets. MT-dependent transport contributes to neuronal development in growing dendrites and axons via slow axonal transport of cytoskeletal elements and, later, it sustains survival and homeostasis of neurons via fast axonal transport of organelles including mitochondria, lysosomes, and vesicles carrying various types of proteins (growth factors, synaptic proteins…). Axonal transport defect is a hallmark of several neurodegenerative disorders, whose disruption affects neuronal function and survival[1–4]. This is exemplified by the loss of activity of the Elongator complex, which is associated with both neurodegeneration and axonal transport defects[5,6]. This molecular complex, conserved from yeast to human, is composed of two copies of six distinct protein subunits (Elp1 to Elp6)[7]. Elp3 is the enzymatic core of the Elongator and harbors two highly conserved sub-domains, a tRNA acetyltransferase[8] and a S-adenosyl methionine binding domains[9]. Elp1 is the scaffolding subunit of the complex, but disruption of any of the Elongator subunits leads to comparable phenotype in eukaryotes, suggesting that all subunits are essential for the integrity and activity of the complex[10,11]. Elongator serves molecular functions in distinct subcellular compartments[12] but predominantly accumulates in the cytoplasm where it promotes the formation of 5-methoxycarbonylmethyl (mcm$^5$) and 5-carbamoylmethyl (ncm$^5$) on side-chains of wobble uridines (U$_{34}$) of selected tRNAs, thereby regulating protein translation[8,13]. Convergent observations support a role for Elongator in intracellular transport in the nervous system as Elp3 is enriched at the presynaptic side of neuromuscular junction buttons in flies, where its expression is required for synapse integrity and activity[14,15]. Elongator subunits are also detected in protein extracts from purified motile vesicles isolated from the mouse cerebral cortex[16], and they colocalize with the vesicular markers SV2 and RAB3A in human embryonic stem cells derived neurons[17]. In humans, mutation of the gene coding for ELP1, underlies Familial dysautonomia (FD), a devastating disease that mostly affects the development and survival of neurons from the autonomic nervous systems[18,19]. Moreover, other neurological disorders are also associated with mutations affecting the expression or the activity of Elongator[5,20–23]. Experimental data on animal models show that interfering with Elongator activity promotes early developmental and progressive neurodegenerative defects, including poor axonal transport and maintenance[24–27]. At the molecular level, loss of Elongator correlates with poor acetylation of α-tubulin lysine 40 (K40) in neuronal microtubules (MT)[6,25,28]. This post-translational modification (PTM) modulates axonal transport by facilitating the recruitment of molecular motors to MTs[29] and the loading of motile vesicles on motors[6]. The acetylation of MTs relies mostly on a vesicular pool of tubulin acetyltransferase 1 (Atat1)[30] that catalyzes the transfer of acetyl groups from Acetyl-Coenzyme-A (Acetyl-CoA) onto lysine 40 (K40) of α-tubulin[31,32]. How loss of Elongator affects MT acetylation, and whether a functional correlation between Elongator and Atat1 exists, remains to be explored.

Here, by combining cellular and molecular analyses in mouse cortical neurons in vitro and fly larva motoneurons (MNs) in vivo, we show that Elongator is expressed in vesicles and modulates axonal transport and acetylation of α-tubulin across species by regulating the stability of ATP-citrate lyase (Acly). This enzyme is enriched in vesicles and produces acetyl-coa, thereby providing acetyl groups for Atat1 activity towards MTs. Importantly, analysis of primary fibroblasts from FD patients show molecular defects comparable to those observed in mice and fly projection neurons depleted in Elongator. Therefore, our data may shine a light on the pathophysiological mechanisms underlying FD and other neurological disorders resulting from impaired Elongator activity.

## Results

**Elongator and Atat1 cooperate in a common pathway to regulate the acetylation of α-tubulin and axonal transport.** In order to understand how Elongator controls tubulin acetylation and MT-based transport, we performed complementary experiments in distinct animal models that lack Elongator activity. We compared the level of α-tubulin acetylation in cultured cortical projection neurons (PN) that were isolated from embryonic (E) day 14.5 WT or Elp3cKO mouse embryos (conditional loss of *Elp3* in cortical progenitors upon breeding *Elp3*lox/lox[33] and FoxG1:Cre[34] transgenic mice). Axon of *Elp3* cKO PNs, as well as cortical extracts from E14.5 Elp3 cKO embryos displayed significant reduction of α-tubulin acetylation (Fig. 1a, Supplementary Fig. 1a). We next assessed axonal transport in PNs that were cultured in microfluidic devices for 5 days and incubated with specific dyes to track lysosomes (LysoTracker®) and mitochondria (MitoTracker®) movements by time-lapse videomicroscopy (Fig. 1b). PNs showed a significant reduction of average and moving velocities (in anterograde and retrograde directions) of lysosomes and mitochondria along axons, which correlated with an increase of their pausing time, as compared to WT PNs (Supplementary Fig. 1b–h). These results confirmed that loss of Elongator activity leads to defects of α-tubulin acetylation and molecular transport in cortical neurons. Since a vesicular pool of Atat1 promotes acetylation of α-tubulin, thereby controls axonal transport in cortical PNs[30], we postulated that Atat1 and Elongator might contribute to axonal transport via a shared molecular pathway. To test this hypothesis, we infected cortical PNs from E14.5 WT and *Atat1* KO mice with lentiviruses expressing either sh-Elp3 or sh-Control (Supplementary Fig. 1i–j), and we cultured them in microfluidics devices for five days (Fig. 1b). Targeting Elp3 in WT neurons impaired lysosomes and mitochondria transport across PN axons. However, we did not observe an additive effect on MT acetylation or axonal transport kinetics upon reduction of Elp3 expression in *Atat1* KO PNs (which show reduced acetylation of α-tubulin) (Fig. 1c–e, Supplementary Fig. 1k–o). Comparable observations were made in vivo in MNs of *Drosophila melanogaster* 3$^{rd}$ instar larvae obtained by crossing UAS:*Elp1* or UAS:*Elp3 RNAi* fly lines with D42:Gal4 (further called *Elp1* KD and *Elp3* KD). The knockdown efficiency of the UAS:*Elp1* and UAS:*Elp3* in postmitotic neurons was validated in adult fly heads (Elav:Gal4 driver; Supplementary Fig. 1p–r). We first confirmed that the level of α-tubulin acetylation was reduced in *Elp1* and *Elp3* KD 3$^{rd}$ instar larvae MNs, a defect genetically rescued by expressing human (h) ELP3 or by co-targeting the main α-tubulin deacetylase Hdac6 with RNAi (*Elp1;Hdac6* KD, *Elp3;Hdac6* KD) (Fig. 1f, Supplementary Fig. 1r). Moreover, knocking down both *Elp3* and *Atat1* (*Elp3;Atat1* KD) did not further reduce the α-tubulin acetylation level, as compared to *Elp3* KD alone (Fig. 1f; *p* = 0.259 and *p* = 0.775, respectively). In vivo time-lapse recording of the synaptotagmin-GFP axonal transport (SYT1-GFP) in MNs of anesthetized fly 3rd instar larvae (Fig. 1g) correlated with levels of α-tubulin acetylation (Fig. 1f). We observed a reduction of both the average and moving velocities of SYT1-GFP vesicles together with an extension of their pausing time both in *Elp1* KD or *Elp3* KD larvae when compared to controls, and with no cumulative defects in *Elp3;Atat1* KD flies (Fig. 1h–j, Supplementary Fig. 1s). Moreover, knocking down *Hdac6* fully rescued the observed axonal transport defects observed in Elp1/3 KD in 3$^{rd}$ instar larvae MNs

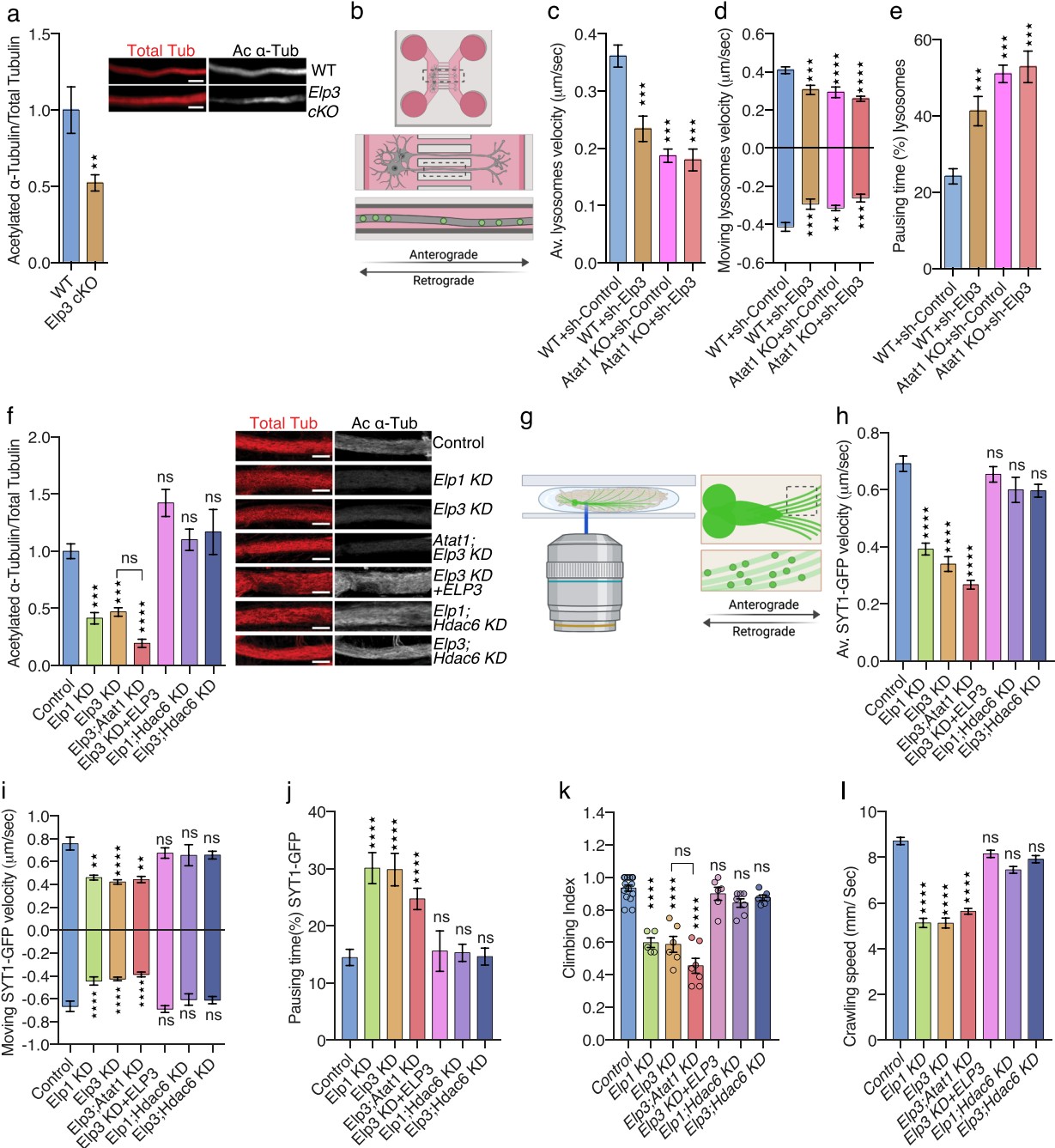

(*Elp1;Hdac6* KD, *Elp3;Hdac6* KD) (Fig. 1h–j, Supplementary Fig. 1s).

Interfering with Elongator activity affects codon-biased translation that can ultimately lead to protein aggregation[35], thereby blocking axonal transport[36]. However, we did not detect significant protein aggregation in the axon of either cultured cortical neurons from *Elp3* cKO mice (Supplementary Fig. 1t) or motoneurons from *Elp1* or *Elp3* KD 3rd instar larvae in vivo (Supplementary Fig. 1u), while axonal aggregates formed upon blocking proteasome activity with MG132 incubation. Altogether, these results suggest that the axonal transport defects observed in Elongator deficient neurons unlikely result from a local accumulation of protein aggregates.

Since impaired axonal transport in fly MNs leads to locomotion defects[1,37], we measured larvae crawling speed and adult flies climbing index as functional readouts of axonal

transport activity in MNs[30,37]. These parameters were affected upon depletion of either *Elp1* or *Elp3* (Fig. 1k–l), and were likely resulting from MT acetylation and axonal transport defects, as they were rescued by the knockdown of *Hdac6* (Fig. 1k–l) and no morphological changes of neuromuscular junction synapses were observed (Supplementary Fig. 1v).

Elongator shares high amino acid similarity across species (Supplementary Fig. 1w). Expression of human *ELP3* in MNs of 3rd instar larvae of *Elp3* KD flies (*Elp3* KD + *ELP3*) rescued α-tubulin acetylation (Fig. 1f), axonal transport parameters (Fig. 1h–j, Supplementary Fig. 1s) and locomotion behavior defects (Fig. 1k–l), indicating that ELP3, and by extension Elongator, has a conserved evolutionary role from human to fly in axonal transport. Moreover, we show that loss of Elongator activity does not worsen the axonal phenotype of *Atat1* KO neurons, further suggesting that these molecules act in a common

**Fig. 1 Reduction of Elongator activity leads to axonal transport defects in mouse and fly. a** Immunolabelings and quantification of acetylated α-tubulin (Ac α-Tub) and total tubulin (Tot Tub) levels in axon of E14.5 WT and *Elp3 cKO* mice cortical neurons cultured for 5 DIV in microfluidic devices. Scale bar is 10 µm. **b** Experimental set up to record axonal transport by time-lapse microscopy in E14.5 mice cortical neurons cultured for 5 DIV in microfluidic devices. **c–e** Histograms showing axonal transport of lysosomes (LysoTracker®) in WT or *Atat1 KO* cultured neurons infected with sh-Control or sh-*Elp3* to analyze average (av.) velocity **c**, moving velocity **d,** and percentage of pausing time **e**. **f–l** Study of *Drosophila melanogaster* expressing RNAi under a motoneuron (MN)-specific driver (D42:GAL4); *Elp1* KD, *Elp3* KD, *Atat1; Elp3* KD, *Elp3* KD + human *ELP3*, *Elp1; Hdac6* KD and *Elp3;Hdac6* KD. **f** Immunolabelings and quantification of acetylated α-tubulin (Ac α-Tub) and total tubulin (Tot Tub) levels in MN axons of 3rd instar larvae, genotype as indicated. Scale bar is 10 µm. **g** Experimental set up for in vivo time-lapse recording of Synaptotagmin-GFP (Syt1-GFP) axonal transport in 3rd instar larvae motoneurons, to analyze average (av.) velocity **h**, moving velocity **i**, and percentage of pausing time **j**. **k–l** Locomotion assays: histograms of the climbing index of adult flies **k** and the crawling speed of 3rd instar larvae **l**. Description of graphical summaries here within are histograms of means ± SEM. Significance was determined by: **a** two-sided *t* test, **c, d, e, f, h, i, j, l** two-sided Kruskal–Wallis test, and **k** two-sided one-way analysis of variance (ANOVA). Specifically, [(**a**) $p = 0.0070$, $t = 2.812$, df = 5; (**c**) $p < 0.0001$ and $K = 80.47$; (**d**) $p < 0.0001$, $K = 34.30$ and $p < 0.0001$, $K = 23.46$ for anterograde and retrograde, respectively, (**e**) $p < 0.0001$ and $K = 73.34$, (**f**) $p < 0.0001$, and $K = 88.16$, (**h**) $p < 0.0001$, $K = 223.6$, (**i**) $p < 0.0001$, $K = 110.3$, and $p < 0.0001$, $K = 45.69$ for anterograde and retrograde, respectively, (**j**)$p < 0.0001$, $K = 46.06$, (**k**) $p < 0.0001$, $F = 28.74$; (**l**) $p < 0.0001$, $K = 86.04$. In addition, the post hoc multiple comparisons, to analyze statistical difference of each condition compared to control for (**c, d, e, f, h, i, j, l**) Dunn's test, for (**k**) Dunnet's test and are **$p < 0.01$, ***$p < 0.001$, and ****$p < 0.0001$. **a** Number of axons: WT $n = 28$; Elp3 cKO $n = 28$. 3 mice per group. **c** Number of vesicles: WT + sh-Control $n = 334$; WT + sh-Elp3 $n = 101$; Atat1 KO + sh-Control $n = 278$; Atat1 KO + sh-Elp3 $n = 79$. 3 mice per group. **d** Number of vesicles: WT + sh-Control $n = 180$ (anterograde), $n = 135$ (retrograde); WT + sh-Elp3 $n = 66$ (anterograde), $n = 56$; Atat1 KO + sh-Control $n = 129$ (anterograde), $n = 163$ (retrograde); Atat1 KO + sh-Elp3 $n = 58$ (anterograde), $n = 41$ (retrograde). 3 mice per group. **e** Number of vesicles: WT + sh-Control $n = 333$; WT + sh-Elp3 $n = 104$; Atat1 KO + sh-Control $n = 278$; Atat1 KO + sh-Elp3 $n = 79$. 3 mice per group. **f** Number of motoneurons: Control $n = 32$; Elp3 KD $n = 20$; Elp1 KD $n = 18$; Elp3;Atat1 KD $n = 19$; Elp3 KD + hELP3 $n = 14$; Elp1;Hdac6 KD $n = 36$; Elp3;Hdac6 KD $n = 15$. 5 larvae per group. **h** Number of vesicles: Control $n = 487$; Elp1 KD $n = 130$; Elp3 KD $n = 217$; Elp3;Atat1 KD $n = 143$; Elp3 KD + ELP3 $n = 397$; Elp1;Hdac6 KD $n = 80$; Elp3;Hdac6 KD $n = 421$. Number of larvae per group: Control: $n = 7$; Elp1 KD, Elp3 KD, Elp3;Atat1 KD, Elp3 KD + ELP3, Elp1;Hdac6 KD, Elp3;Hdac6 KD $n = 12$. **i** Number of vesicles: Control $n = 127$ (anterograde), $n = 159$ (retrograde); Elp1 KD $n = 118$ (anterograde), $n = 105$ (retrograde); Elp3 KD $n = 194$ (anterograde), $n = 203$ (retrograde); Elp3;Atat1 KD $n = 68$ (anterograde), $n = 81$ (retrograde); Elp3 KD + ELP3 $n = 173$ (anterograde), $n = 247$ (retrograde); Elp1;Hdac6 KD $n = 50$ (anterograde), $n = 39$ (retrograde); Elp3;Hdac6 KD $n = 283$ (anterograde), $n = 264$ (retrograde). Number of larvae per group: Control: $n = 7$; Elp1 KD, Elp3 KD, Elp3;Atat1 KD, Elp3 KD + ELP3, Elp1;Hdac6 KD, Elp3;Hdac6 KD $n = 12$. **j** Number of vesicles: Control $n = 488$; Elp1 KD $n = 217$; Elp3 KD $n = 130$; Elp3;Atat1 KD $n = 429$; Elp3 KD + ELP3 $n = 90$; Elp1;Hdac6 KD $n = 427$; Elp3;Hdac6 KD $n = 397$. Number of larvae per group: Control: $n = 7$; Elp1 KD, Elp3 KD, Elp3;Atat1 KD, Elp3 KD + ELP3, Elp1;Hdac6 KD, Elp3;Hdac6 KD $n = 12$. **k** Number of vials: Control $n = 15$; Elp1 KD $n = 5$; Elp3 KD $n = 6$; Elp3;Atat1 KD $n = 7$; Elp3 KD + ELP3 $n = 6$; Elp1;Hdac6 KD $n = 7$; Elp3;Hdac6 KD $n = 6$. Each vial contains 10 adult flies. **l** Larvae per group: Control $n = 21$; Elp1 KD $n = 15$; Elp3 KD $n = 12$; Elp3;Atat1 KD $n = 18$; Elp3 KD + ELP3 $n = 17$; Elp1;Hdac6 KD $n = 12$; Elp3;Hdac6 KD $n = 12$. Source data are provided with this paper.

molecular pathway to modulate MT acetylation and axonal transport[31,32].

**Loss of Elongator impairs the production of acetyl-CoA, thereby interfering with Atat1- mediated MT acetylation.** Since *Hdac6* knockdown rescued the axonal transport deficits observed in neurons lacking Elongator activity (Fig. 1h–j), we tested whether these defects may arise from a change of expression or activity of Hdac6 and/or Atat1, the enzymes that control α-tubulin de/acetylation, respectively[38,39]. Since no change in Hdac6 and Atat1 (isoform #3 and #4)[30] expression in cortical extracts from *Elp3* cKO and WT littermate newborn mice (Fig. 2a–c) were observed, we measured the activity of these enzymes. The deacetylation activity of Hdac6 toward MTs, assessed in vitro by incubating free acetylated α-tubulin from bovine brains with cortical extracts from newborn *Elp3* cKO or WT littermate controls[38], was comparable between conditions (Fig. 2d). Atat1 activity was measured by using an in vitro α-tubulin acetylation assay[30]. For this assay, pre-polymerized unacetylated MTs from HeLa cells were incubated with acetyl-CoA and P0 mouse brain extracts from *Elp3* cKO or WT littermate controls (Fig. 2e) to assess the level of acetylation of α-tubulin. In striking contrast to brain extracts from *Atat1* KO mouse pups, we did not observe any differences in MT acetylation levels between the cortical extracts from *Elp3* cKO and WT mice upon addition of acetyl-CoA (Fig. 2e). Altogether, these results suggest that loss of Elongator activity: i) does not impair α-tubulin acetylation by changing the expression or activity of Hdac6 or Atat1; ii) may interfere with the metabolic production of acetyl-CoA. Accordingly, western blotting of E14.5 Elp3cKO cortical extracts showed that expression of Acyl-coenzyme A synthetase short-chain family member (Acss2) and Acly, the

enzymes that generate acetyl-CoA[40], were significantly reduced, as compared to control extracts (Fig. 2f–h). Since α-tubulin acetylation is mainly driven by a vesicular pool of Atat1 in axons[30], we performed sub-fractionation of cortical lysates (Fig. 2i; T: total, P1: nuclei, S2: cytosolic + vesicular, P2: membranes, S3: cytosolic, P3: vesicular) from adult WT mice to analyze the subcellular distribution of Acss2 and Acly. While Acss2 showed a strict cytosolic distribution (S3), we found that Acly was strongly enriched together with Elp1 and Elp3 in vesicles (P3) (Fig. 2j–l). These results were supported by immunolabeling of cortical neurons that show expression of Acly in some synapthophysin + vesicles and Lamp1 + late endosomes in PN axons (Fig. 2m–n). Moreover, proteomic analysis of cortical extracts from newborn mice confirmed the vesicular enrichment of several Elongator subunits, Atat1, Acly, but not Acss2 (Supplementary Fig. 2a). Altogether, these data suggest that α-tubulin acetylation could be regulated by homeostatic changes in Acly-dependent acetyl-CoA concentration, as previously observed for histone acetylation[41]. Specifically, we postulated that loss of Elongator could interfere with MT acetylation and axonal transport via reduction of Acly expression. To test this hypothesis, we performed an in vitro acetyl-CoA production assay (malate dehydrogenase coupled assay)[41], where subcellular fractions of cortical brain extracts from *Elp3* cKO and WT littermate newborns were mixed with Acly substrates (ATP, CoA, and citrate) (Fig. 2o). The vesicular fraction (P3), which is strongly enriched into Elongator subunits (Elp1 and Elp3), Atat1 and Acly (Fig. 2j, Supplementary Fig. 2a), more efficiently produced acetyl-coA (Fig. 2o–p). While cortical extracts from *Elp3* cKO E14.5 mouse embryos showed no change of *Acly* transcription (Fig. 2q), we observed a reduction of Acly expression (Fig. 2f–g) together with a correlative decrease of MTs acetylation (Supplementary

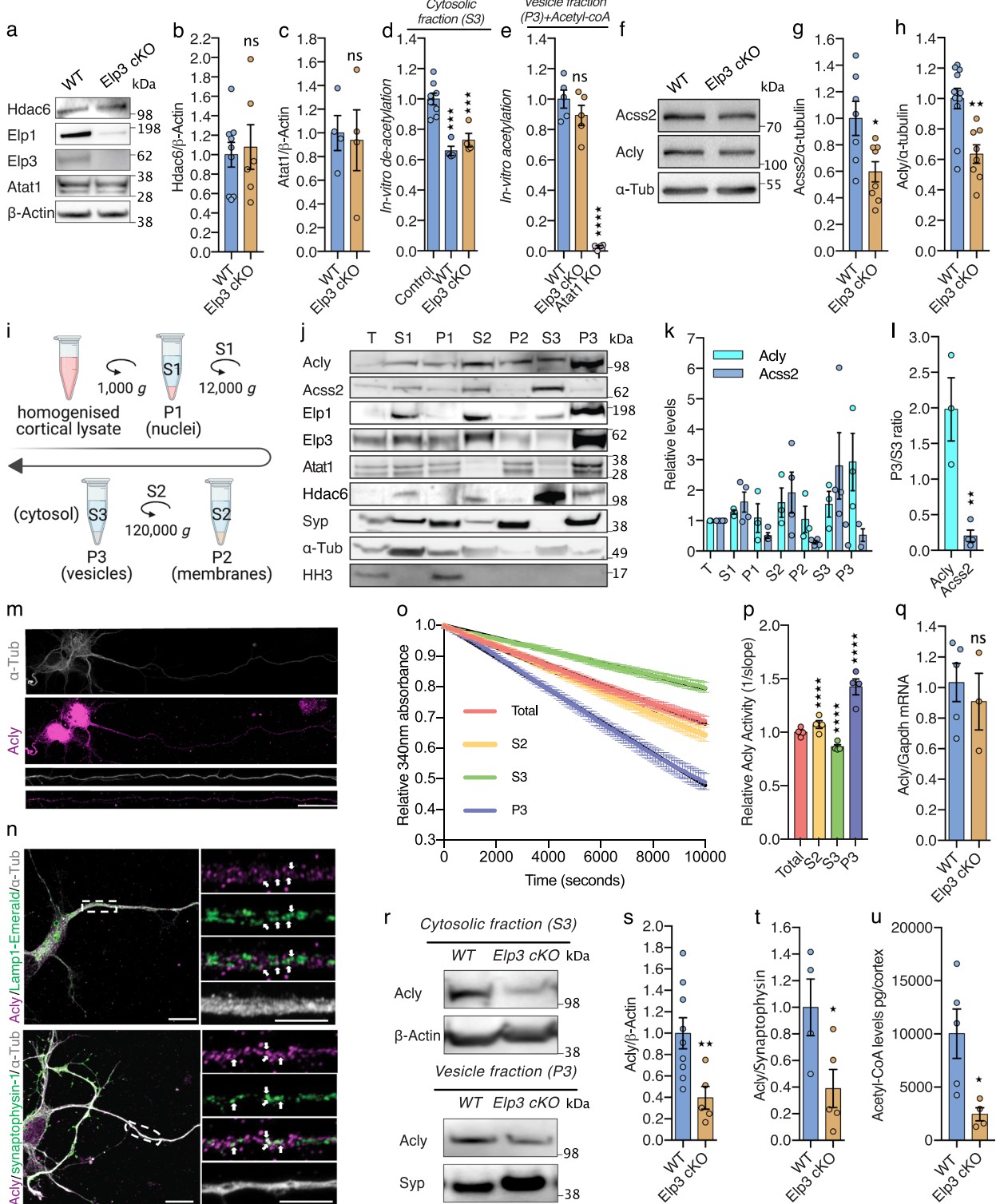

Fig. 1a). Indeed, the analysis of *Elp3* cKO and WT newborn cortices show a robust decrease of Acly protein level in both the cytosolic and the vesicular fractions (*P3*, Fig. 2r–t). Comparable reduction of Acly (but not Atat1) was detected by mass spectrometry in the vesiculome of P0 Elp1 KO brains (Supplementary Fig. 2b–d). Moreover, we detected a reduction of acetyl-CoA level in cortical extracts from *Elp3* cKO mice, as compared to WT newborn littermates by using LC-MS metabolic analysis (Fig. 2u). Altogether, these results support a molecular cooperation occurring at vesicles between the Elongator and Atat1 to drive

MT acetylation via the regulated expression of the enzyme Acly, which provides acetyl-group donors to Atat1 for catalyzing α-tubulin acetylation in neurons.

**Expression of Acly rescues both acetylation of α-tubulin and axonal transport defects in flies and mice that lack Elongator activity.** In order to test whether Acly is required for proper axonal transport, we incubated WT mouse PNs with the Acly inhibitor hydroxycitric acid (HCA)[42]. This treatment reduced the level of α-tubulin acetylation in PNs axons (Fig. 3a). Moreover, incubation of

**Fig. 2 Elongator deletion impairs tubulin acetylation and axonal transport via reduction of Acly-dependent acetyl-coA production. a–c** Western blotting to detect and quantify Hdac6, Elp1, Elp3, Atat1 (isoform #3 and #4)[30], and ß-Actin in cortical extracts from newborn WT and *Elp3 cKO* mice. Histograms of proportion of Hdac6 **b** and Atat1 **c** expression to ß-Actin. **d** In vitro deacetylation assay of endogenously acetylated bovine brain tubulin incubated for 4 h with extracts (S3) of cortices isolated from WT, *Elp3 cKO*, or without tissue extract (control). **e** In vitro acetylation assay of nonacetylated MTs from HeLa cells incubated for 2 h with Acetyl-CoA and extracts (P3) of brain cortices isolated from WT, *Elp3 cKO* or *Atat1 KO* mice. **f–h** Immunoblotting to detect Acss2, Acly, and α-Tub cortical extracts from E14.5 WT and *Elp3 cKO* embryos and histograms of proportion of Acss2 (**g**) and Acly (**h**) expression to α-Tub. **i** Experimental pipeline for subcellular fractionation of P0 mouse cortical extracts: T, total; S1, post-nuclear; P1, nuclear; S2, cytosol and vesicles; P2, large membranes; S3, cytosol; and P3, vesicles. **j–l** Subcellular fractionation (T, S1, P1, S2, P2, S3, P3) of mouse brain cortex immunoblotted with specific antibodies to detect Acss2, Acly, Elp1, Elp3, Atat1 (isoform #3 and #4)[30], Hdac6, synaptophysin (Syp), α-tubulin (α-Tub), and Histone H3 (HH3). **k–l** Histograms of proportion of Acss2 and Acly across subcellular fractions **k** or expressed as P3/S3 ratio **l**. Immunolabelings of E14.5 mice cortical neurons cultured for 5 DIV to detect Acly (purple) and α-tubulin (α-Tub, grey) **m**; Acly, Synaptophysin-1 (green) and α-Tub (**n**, top); Acly, Lamp-1-Emerald (green) and α-Tub (**n**, bottom). Scale bars are 20 μm (m), 10 μm (n). **o–p** Analysis of Acly activity by malate dehydrogenase coupled method performed in WT and *Elp3 cKO* brain cortex lysates. Histogram of relative Acly activity over assay (**o**) and of slopes (**p**) from the linear phase of the reaction. **q** qRT-PCR analysis of *Acly* mRNA in cortical brain extracts of *Elp3 cKO* or WT littermate mice. **r–t** Immunoblots and quantification of cytosolic (S3, **s**) and vesicular (P3, **t**) fractions of Acly protein from newborn WT and *Elp3 cKO* mice brain cortices. **u** LC-MS quantification of Acetyl-CoA levels in WT and *Elp3 cKO* P0 mice brain cortex lysates. Description of graphical summaries here within are histograms of means ± SEM. Significance was determined by: **b, c, g, h, l, s, t** two-sided *t* test, **q, u** two-sided Mann–Whitney test, **d, e, p** two-sided one-way analysis of variance (ANOVA), and **o** two-sided two-way ANOVA. Specifically, [(**b**) $p = 0.8411$, $t = 0.2093$, $df = 6$; (**c**) $p = 0.7505$, $t = 0.3248$, $df = 13$; (**d**) $p < 0.0001$, $F = 21.30$; (**e**) $p < 0.0001$, $F = 85.29$; (**g**) $p = 0.0157$, $t = 0.2776$, $df = 13$; (**h**) $p = 0.0013$, $t = 3.861$, $df = 17$; (**l**) $p = 0.0056$, $t = 4.638$, $df = 5$; (**o**) $p < 0.0001$, $F_{Interaction}$ (294, 882) = 68.07; (**p**) $p < 0.0001$, $F = 1352$; (**q**) $p = 0.5714$, $U = 5$; (**s**) $p = 0.0145$, $t = 2.855$, $df = 12$; (**t**) $p = 0.0428$, $t = 2.470$, $df = 7$; (**u**) $p = 0.0159$, $U = 0$. In addition, the post hoc multiple comparisons, to analyze statistical difference` of each condition compared to control for (**d, e, p**) are Holm-Sidak test, and are $*p < 0.05$, $**p < 0.01$, $***p < 0.001$, and $****p < 0.0001$. Number of mice: (**b**) WT $n = 9$; Elp3 cKO $n = 6$; (**c**) WT $n = 4$; Elp3 cKO $n = 4$; (**d**) Control $n = 8$; WT $n = 4$; cKO $n = 4$; (**e**) WT $n = 5$; Elp3 cKO $n = 5$; Atat1 KO $n = 4$; (**g**) WT $n = 7$; Elp3 cKO $n = 8$; (**h**) WT $n = 11$; Elp3 cKO $n = 8$; (**l**) WT $n = 3$ (Acly); WT $n = 4$ (ACSS2); (**o–p**) WT $n = 4$ for 99 time points; (**q**) WT $n = 5$; Elp3 cKO $n = 3$; (**s**) WT $n = 9$; Elp3 cKO $n = 5$; (**t**) WT $n = 4$; Elp3 cKO $n = 5$; (**u**) WT $n = 5$; Elp3 cKO $n = 4$. **e** $p = 0.031$, $t = 2.407$, $df = 13$. Source data are provided with this paper.

HCA reduced average and retrograde moving velocities of lysosomes along axons, which correlated with an increase of their pausing times, as compared to vehicle treatment (Fig. 3b–d, Supplementary Fig. 3a). We next analyzed 3rd instar larvae from *Acly* KD flies[43] whose MNs displayed reduced α-tubulin acetylation (Fig. 3e) together with impairment of SYT1-GFP vesicles transport (Fig. 3f–h, Supplementary Fig. 3b). Knocking down *Hdac6* fully rescued the level of α-tubulin acetylation (Fig. 3e) and the axonal transport defects observed in *Acly* KD MNs (*Acly* KD;*Hdac6* KD, *Acly*;*Hdac6* KD) (Fig. 3f–h, Supplementary Fig. 3b). We next postulated that the impairment of α-tubulin acetylation and axonal transport observed upon loss of Elongator (Supplementary Fig. 1a–h) may arise from the reduced expression of Acly in PNs. To test this hypothesis, we expressed human ACLY (which shares high amino acid homology with its murine and fly homologues; Supplementary Fig. 3c) in E14.5 control and Elp3cKO PNs that were cultured in microfluidic devices for 5DIV. We showed that the level of MT acetylation (Fig. 3i) and the lysosomes and mitochondria transport defects were rescued upon expression of ACLY (Fig. 3j–l, Supplementary Fig. 3d–g). Comparable data were obtained with *Elp3* KD flies (that also show reduction of Acly level in the brain, Supplementary Fig. 3h) that express Acly (UAS:Elp3 RNAi and UAS:Acly, *Elp3* KD + Acly) in MNs during development (3rd instar larvae), where both acetylation of α-tubulin and axonal transport of SYT1-GFP vesicles were rescued to control levels (Fig. 3m–p, Supplementary Fig. 3i). These data were also correlated with improvement of locomotion activities of *Elp3* KD 3rd instar larvae and adult flies upon Acly expression (Supplementary Fig. 3j–k).

Altogether, these results suggest that reduction of Acly expression leads to MTs acetylation and axonal transport defects in Elongator deficient neurons.

**Fibroblasts from Familial Dysautonomia patients show defects of α-tubulin acetylation and MT-dependent transport**. In order to place our findings in a human pathological context, we analyzed primary skin fibroblasts from FD patients carrying the splice site IVS20 + 6T_C variant in *ELP1*[44] that expressed barely detectable amount of ELP1 proteins and low level of ELP3,

resulting from its instability upon nonassembly of the Elongator complex (Supplementary Fig. 4a)[25]. We observed a reduction of acetylation of α-tubulin and MT-dependent transport defects in FD fibroblasts (Supplementary Fig. 4b–g), which were comparable to those observed in neurons from Elongator loss of function fly and mouse models. This modification was not resulting from an increased deacetylation activity of HDAC6 toward MTs in extracts fibroblasts nor an intracellular protein aggregation that would affect MT-dependent transport in FD fibroblasts (Supplementary Fig. 4h–i). Both the reduction of α-tubulin acetylation and the defects of MT-dependent transport in cultured FD fibroblasts were rescued by blocking tubulin deacetylation with the HDAC6 specific inhibitor, Tubastatin (TBA; 10uM during 30 min) (Supplementary Fig. 4c–g). Similarly to the animal models, FD fibroblasts express reduced amount of ACLY proteins (Fig. 4a) and incubation with HCA for 8 h interfered in a dose-dependent manner with MT acetylation (Fig. 4b), in both control and FD fibroblasts, in line with the residual expression of ACLY observed in FD fibroblasts (Fig. 4a). We next showed that over-expression of ACLY in FD fibroblasts rescued the level of acetylation of α-tubulin (Fig. 4c), as well as the defects of MT-dependent transport of lysosomes (Fig. 4d–f, Supplementary Fig. 4j). Altogether, these results demonstrate that reduction of ACLY in FD fibroblasts leads to MT acetylation and MT-dependent transport impairments; these molecular defects likely contribute to the pathological mechanisms underlying FD.

**Loss of elongator activity interferes with Acly stability**. Elongator is a tRNA modifier and its loss of activity may directly impair the translation of *Acly* mRNAs. To address this question, we performed a polysomal fractionation of cortical extracts of P0 Elp3 cKO and WT littermates to analyze the association of *Acly* mRNAs with ribosomes[45] and found no differences between genotypes (Fig. 5a–b). These data were confirmed by labeling cultured Elp3 cKO an WT PNs with puromycin for 4 min, a tRNA analogue that incorporates into nascent peptides, causing their premature chain termination (reviewed in[46]). We visualized the de novo synthesis of Acly by combining anti-puromycin with

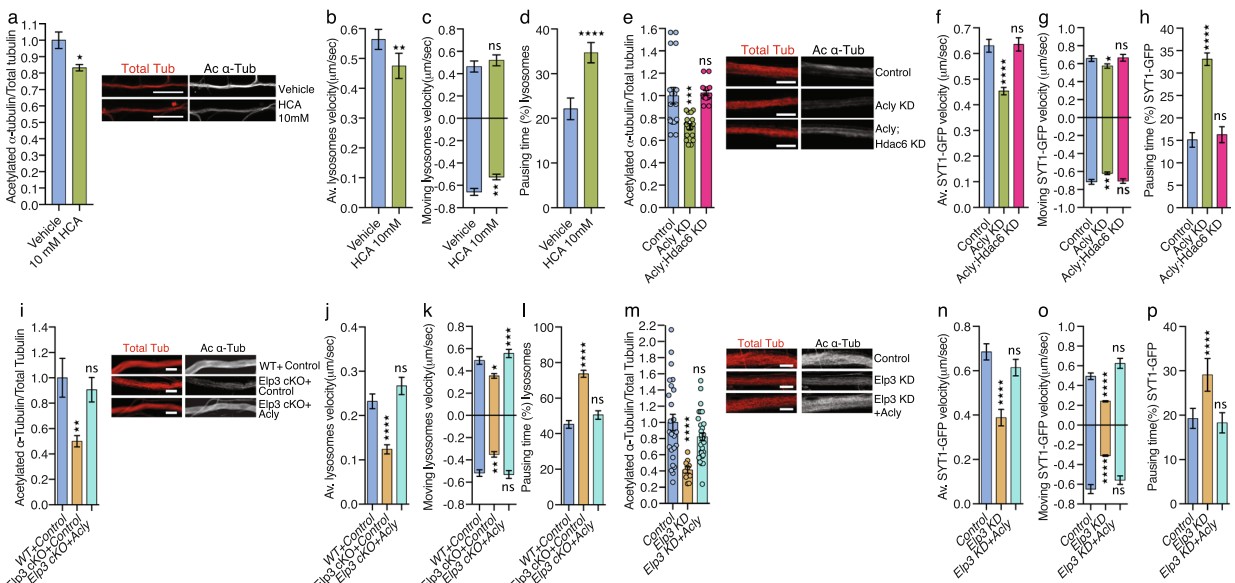

**Fig. 3 Acly/ACLY expression rescues α-tubulin acetylation and molecular transport defects upon loss of Elongator activity in mouse and fly neurons.**
**a** Immunolabeling and quantification of acetylated α-tubulin (Ac α-Tub) and total tubulin (Tot α-Tub) levels in axons of DIV5 cultured cortical PNs isolated from E14.5 embryos and incubated with 10 mM hydroxy-citrate (HCA) during 8 h prior to fixation. Scale bar is 10 µm. **b–d** Time-lapse recording of axonal transport in corresponding PNs cultured for 5 DIV in microfluidic devices and treated with vehicle- or HCA-supplemented medium for 8 h showing (av.) velocity **b**, moving velocity **c**, and percentage of pausing time **d** of lysosomes (LysoTracker®). **e** Immunolabeling and quantification of acetylated α-tubulin (Ac α-Tub) and total tubulin (Tot α-Tub) levels in MN axons from 3rd instar larvae of *Drosophila melanogaster* control or expressing Acly RNAi (Acly KD) under a MN-specific driver (D42:GAL4). Scale bar is 10 µm. **f–h** Time-lapse recording of Synaptotagmin-GFP (Syt1-GFP) axonal transport in 3rd instar larvae MNs of control or *Acly* KD to analyze average (av.) velocity **f**, moving velocity **g**, and percentage of pausing time **h**. **i** Immunolabeling and quantification of acetylated α-tubulin (Ac α-Tub) and total tubulin (Tot α-Tub) levels in axons of cultured cortical PNs isolated from E14.5 WT and *Elp3cKO* embryos transfected with Acly or control. Scale bar is 10 µm. **j–l** Cortical neurons isolated from E14.5 WT and Elp3cKO embryos were transfected with control or ACLY expressing constructs and cultured for 5 DIV in microfluidic devices to perform time-lapse recording of axonal transport and measure average (av.) velocity **j**, moving velocity **k**, and percentage of pausing time **l** of lysosomes (LysoTracker®). **m** Immunolabeling and quantification of acetylated α-tubulin (Ac α-Tub) and total tubulin (Tot α-Tub) levels in MN axons of 3rd instar larvae: control, *Elp3* KD, and *Elp3* KD + human *ACLY*. Scale bar is 10 µm. **n–p** In vivo live imaging and behavior measurements in 3rd instar larva: control, *Elp3* KD, and *Elp3* KD + human *ACLY*. **n–p** Time-lapse recording of axonal transport of Synaptotagmin-GFP (Syt1-GFP) in MNs to analyze average (av.) velocity **n**, moving velocity **o**, and percentage of pausing time **p**. Description of graphical summaries here within are histograms of means ± SEM. Significance was determined by: **a** two-sided *t* test, **b–d** two-sided Mann–Whitney test, **i** two-sided one-way analysis of variance (ANOVA), and **e, f, g, h, j, k, l** two-sided Kruskal–Wallis one-way ANOVA. Specifically, [(**a**) $p = 0.0166$, $t = 2.445$, df = 81; (**b**) $p = 0.0039$, $U = 6807$; (**c**) $p = 0.2524$, $U = 2152$ and $p = 0.0032$, $U = 4559$ for anterograde and retrograde, respectively; (**d**) $p < 0.0001$, $U = 6201$; (**e**) $p < 0.0001$, $K = 26.17$; (**f**) $p < 0.0001$, $K = 55.55$; (**g**) $p = 0.0130$, $K = 8.69$ and $p = 0.0004$, $K = 15.55$ for anterograde and retrograde, respectively; (**h**) $p < 0.0001$, $K = 93.1$; (**i**) $p = 0.0042$, $F = 4.969$; (**j**) $p < 0.0001$, $K = 28.69$; (**k**) $p = 0.001$, $K = 13.85$ and $p < 0.0001$, $K = 251.5$; (**l**) $p < 0.0001$, $K = 24.74$; (**m**) $p = 0.0003$, $F = 9.158$; (**n**) $p < 0.0001$, $K = 21.66$; (**o**) $p < 0.0001$, $F = 25.90$, and $p < 0.0001$, $F = 20.96$ for anterograde and retrograde, respectively; (**p**) $p < 0.0001$, K = 22.35. In addition, the post hoc multiple comparisons, to analyze statistical difference of each condition compared to control for (**e**) Kruskal–Wallis test, for (**i, m**) is Dunnet's test, for (**f, g, h, j, l, n, o, p**) are Sidak's test and are *$p < 0.05$, **$p < 0.01$, ***$p < 0.001$, and ****$p < 0.0001$. **a** Number of axons: Vehicle $n = 88$; 10 mM HCA $n = 151$; 2 mice per group. **b**) Number of axons: Vehicle $n = 133$; 10 mM HCA $n = 129$. 2 mice per group. **c** Number of vesicles: Vehicle $n = 63$ (anterograde), $n = 113$ (retrograde); 10 mM HCA $n = 77$ (anterograde), $n = 105$ (retrograde). d Number of vesicles: Vehicle $n = 133$; 10 mM HCA $n = 129$. **e** Number of axons: Control $n = 9$; Acly KD n $= 9$; Acly;Hdac6 KD $n = 9$. 6 larvae per group. **f** Number of vesicles: Control $n = 234$; Acly KD $n = 422$; Acly;Hdac6 KD $n = 182$. 3 animals per group. **g** Number of vesicles: Control $n = 131$ (anterograde), $n = 134$ (retrograde); Acly KD $n = 195$ (anterograde), $n = 337$ (retrograde); Acly;Hdac6 KD $n = 89$ (anterograde), $n = 119$ (retrograde). **h** Number of vesicles: Control $n = 234$; Acly KD $n = 422$; Acly;Hdac6 KD $n = 182$. **i** Number of axons: WT + Control $n = 28$; Elp3 cKO + Control $n = 25$; Elp3 cKO + Acly $n = 23$. Number of mice per group: WT + Control $n = 5$; Elp3 cKO + Control $n = 3$; Elp3 cKO + Acly $n = 5$. **j** Number of vesicles: WT + Control $n = 441$; Elp3 cKO + Control $n = 429$; Elp3 cKO + Acly $n = 385$. Number of mice per group: WT + Control $n = 5$; Elp3 cKO + Control $n = 3$; Elp3 cKO + Acly $n = 5$. **k** Number of vesicles: WT + Control $n = 108$ (anterograde), $n = 143$ (retrograde); Elp3 cKO + Control $n = 111$ (anterograde), $n = 79$ (retrograde); Elp3 cKO + Acly $n = 103$ (anterograde), $n = 107$ (retrograde). Number of mice per group: WT + Control $n = 5$; Elp3 cKO + Control $n = 3$; Elp3 cKO + Acly $n = 5$. **l** Number of vesicles: WT + Control $n = 441$; Elp3 cKO + Control $n = 429$; Elp3 cKO + Acly $n = 385$. Number of mice per group: WT + Control $n = 5$; Elp3 cKO + Control $n = 3$; Elp3 cKO + Acly $n = 5$. **m** Number of axons: Control $n = 25$; Elp3 KD $n = 10$; Elp3 KD + Acly $n = 37$. 3 mice per group. **n** Number of vesicles: Control $n = 304$; Elp3 KD $n = 78$; Elp3 KD + Acly $n = 181$. Number of larvae per group: Control $n = 5$; Elp3 KD $n = 10$; Elp3 KD + Acly $n = 10$. **o** Number of vesicles: Control $n = 87$ (anterograde), $n = 141$ (retrograde); Elp3 KD $n = 99$ (anterograde), $n = 141$ (retrograde); Elp3 KD + Acly $n = 80$ (anterograde); $n = 81$ (retrograde). Number of larvae per group: Control $n = 5$; Elp3 KD $n = 10$; Elp3 KD + Acly $n = 10$. **p** Number of vesicles: Control $n = 252$; Elp3 KD $n = 84$; Elp3 KD + Acly $n = 234$. Source data are provided with this paper.

anti-Acly antibodies and performing proximity ligation assay (PLA) to reveal Acly puro-PLA puncta in PNs (Fig. 5c–d). We showed comparable somatic and axonal Acly puro-PLA puncta in PNs from both genotypes (Fig. 5e–f). Together, these data suggest that loss of Elongator activity does not interfere with the translation of *Acly* mRNAs in cortical PNs.

We next tested whether the stability of ACLY might be impaired upon loss of Elongator activity. For this purpose, we cultured

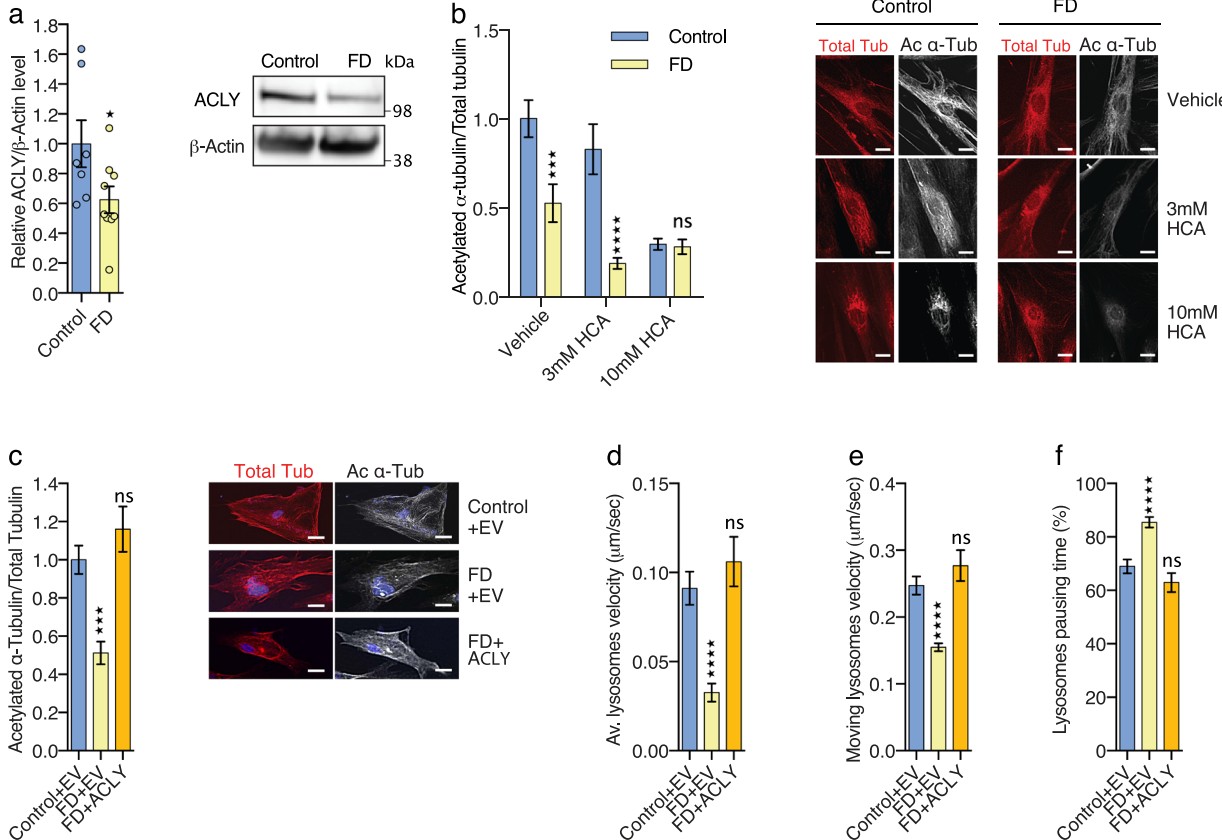

**Fig. 4 ACLY expression rescues defects of α-tubulin acetylation and microtubule-dependent transport in human FD fibroblasts. a** Immunoblotting of ACLY and ß-ACTIN in cultured human primary fibroblast extracts from Control and FD patients. **b** Immunolabeling and quantification of acetylated α-tubulin and total tubulin levels in primary fibroblasts from healthy controls and FD patients incubated with the ACLY inhibitor hydroxy-citrate acid (HCA). Scale bar is 50 μm. **c** Immunolabeling and quantification of acetylated α-tubulin and total α-tubulin levels in extracts from primary fibroblasts from healthy controls and FD patients transfected with control or ACLY expressing plasmids. Scale bar is 50 μm. **d–f** Time-lapse recording of MT-dependent transport of lysosome (LysoTracker®) in fibroblasts from Control or FD patients, transfected with Control or ACLY plasmids, to analyze average (av.) velocity (**d**), moving velocity (**e**) and percentage of pausing time (**f**). Description of graphical summaries here within are histograms of means ± SEM. Significance was determined by: (**a**) two-sided *t* test, (**b**) two-sided two-way analysis of variance (ANOVA), (**c**) two-sided ANOVA, and (**d, e, f**) two-sided Kruskal–Wallis one-way ANOVA. Specifically, [(**a**) $p = 0.0467$, $t = 2.181$, df $= 14$; (**b**) $p = 0.0038$, $F_{interaction}$ (2, 109) $= 5.876$; (**c**) $p < 0.0001$, $F = 13.17$; (**d**) $p < 0.0001$, $K = 48.95$; (**e**) $p < 0.0001$, $K = 35.10$; (**f**) $p < 0.0001$, $K = 40.98$. In addition, the post hoc multiple comparisons to analyze statistical difference of each condition compared to control for (**b**) is Sidak's test, for (**c**) is Dunnet's test, for (**d, e, f**) are Dunn's test, and are *$p < 0.05$, ***$p < 0.001$, and ****$p < 0.0001$. **a** 5 human primary fibroblast lines per group. **b** Number of cells: Control $n = 26$ (Vehicle), $n = 18$ (3 mM HCA), $n = 20$ (10 mM HCA); FD $n = 18$ (Vehicle), $n = 15$ (3 mM HCA), $n = 18$ (10 mM HCA). Number of fibroblast lines per group: Control $n = 5$; FD $n = 4$. **c** Number of cells: Control + EV $n = 15$; FD + EV $n = 15$; FD + Acly $n = 15$. **d** Number of vesicles: Control + EV $n = 262$; FD + EV $n = 217$; FD + Acly $n = 143$. 5 human primary fibroblast lines per group. **e** Number of vesicles: Control + EV $n = 133$; FD + EV $n = 68$; FD + Acly $n = 89$. 5 human primary fibroblast lines per group. **f** Number of vesicles: Control + EV $n = 251$; FD + EV $n = 217$; FD + Acly $n = 143$. 5 human primary fibroblast lines per group. Source data are provided with this paper.

patient fibroblasts and performed cycloheximide chase assay to measure ACLY half-life in several lines of control and FD fibroblasts. We showed a reduction of ACLY stability in FD fibroblasts as compared to control lines (Fig. 5g–h), a phenotype that may result from the disruption of an interaction between the Elongator and Acly, as suggested by co-immunoprecipitation experiments performed in HEK293 cells (Supplementary Fig. 5a). Altogether, these data suggest that loss of Elongator activity reduces Acly/ACLY stability, thereby affecting the local production of acetyl-CoA required by ATAT1 to promote MT acetylation. Such regulation is likely to take place in motile vesicles (Fig. 5i).

## Discussion
Here, we show that loss of Elongator activity impairs both the MT acetylation and bidirectional axonal transport in mouse cortical neurons in culture, as well as in fly motoneurons in vivo,

ultimately resulting in locomotion deficits in flies. These defects are similar to those observed upon loss of Atat1 expression, the enzyme that catalyzes the acetylation of α-tubulin K40 residues in MTs. This shared phenotype prompted us to investigate a possible coordination between the Elongator and Atat1 in the control of MT-dependent axonal transport via α-tubulin acetylation. By combining conditional loss of function models with genetic experiments, we observed a molecular cooperation between Elongator and Atat1 and identified Acly as the common denominator, which provides acetyl groups from Acetyl-CoA to vesicular Atat1 and whose expression depends on Elongator activity. We showed that acute knockdown or pharmacological blockade of Acly impairs MT acetylation and axonal transport in fly and mouse, respectively. Here, the lower effect of HCA treatment on lysosome axonal transport may reflect a sub-efficient reduction of Acetyl CoA availability in mouse PNs, as

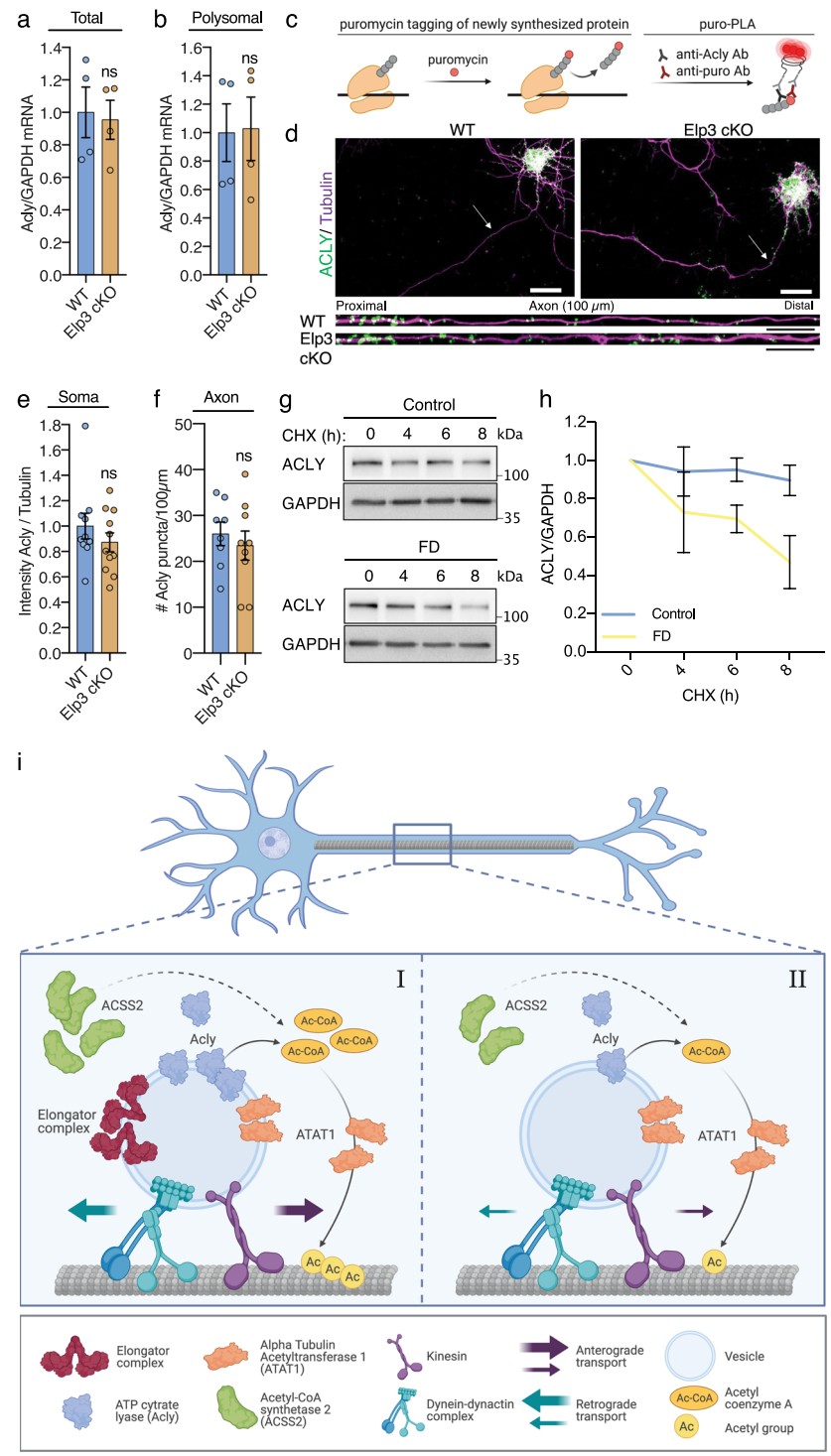

compared to the one obtained in the genetic fly model. On the other hand, raising Acly level not only rescued axonal transport defects in both mouse and fly Elongator models but also improved locomotion activity of *Elp3* KD larvae and flies. We found that loss of Elongator activity does not affect the transcription and translation of *Acly* but rather impairs Acly/ACLY stability. Our in vitro data suggest that Elongator may regulate Acly/ACLY protein stability via molecular interaction, as previously reported for other cytoplasmic proteins[37] (Supplementary Fig. 5). Another non-mutually exclusive mechanism that may act downstream Elongator would involve the regulation of the acetyltransferase[47] P300/CBP-associated factor (PCAF),

which is known to increase Acly stability by promoting its acetylation[47].

We previously showed that motile vesicles are the main driver of MT acetylation in axons[30] and Elongator subunits and Atat1 have been detected in protein extracts from purified motile vesicles[16,30]. Here, we found that the vesicular fraction isolated from mouse cerebral cortical extracts does not only express Elongator subunits and Atat1, but it is also enriched with a functional pool of Acly (Fig. 2j–l). Therefore, we postulate that an Elongator/Acly/Atat1 (EAA) signaling pathway may directly act at vesicles to promote MT acetylation, further supporting their non-canonical role as a platform for local signaling and for

**Fig. 5 Elongator depletion reduces the stability but not translation of Acly. a–b** Transcriptional and translational analysis of Acly mRNA in cortex lysate from newborn WT and Elp3 cKO mice. qPCR quantification of Acly mRNA normalized to GAPDH obtained from lysed cortex (total) (**a**) or from polysomal fraction (polysomal) enriched by sucrose gradient ultracentrifugation of cytoplasmic cortical lysate (**b**) (values are means ± SEM; unpaired *t* test). **c–f** Analysis of local translation events of Acly in cultured cortical PNs from E14.5 WT and Elp3cKO embryos. Schematic of the puro-PLA principle to visualize newly synthesized peptides (**c**). Representative images of newly synthesized Acly in cortical PNs cultured five days in vitro and treated 4 min with puromycin before fixation and processing for puro-PLA; puro-Acly PLA puncta (green), α-tubulin (purple) (**d**). Quantification of somal puro-PLA fluorescence intensity normalized to α-tubulin (**e**) and axonal number of puro-PLA puncta over 100 μm segments (**f**) (values are means ± SEM; unpaired *t* test). Scale bar is 20 μm (top panels) and 10 μm (bottom panels). **g–h** Immunoblotting and quantification of ACLY and GAPDH in human primary fibroblast extracts from Control and FD patients incubated with cycloheximide (CHX, 50 μg/mL) for 0, 4, 6, and 8 h. **i** Summary scheme showing a central role played by Acly in the control of both α-tubulin acetylation and microtubule-dependent transport, which are impaired upon loss of Elongator activity. Description of graphical summaries here within are histograms of means ± SEM. Significance was determined by: (**a, b, e, f**) two-sided *t* test, (**e**) two-sided two-way ANOVA. Specifically, [(**a**) $p = 0.8244$, $t = 0.2318$, df = 6; (**b**) $p = 0.9303$, $t = 0.9114$, df = 6; (**e**) $p = 0.3160$, $t = 1.03$, df = 19; (**f**) $p = 0.5484$, $t = 0.6141$, df = 15; (**h**) $p < 0.0034$, $F_{Interaction} (1, 22) = 10.78$). (**a, b**) 4 animals per group. **e** Number of axons: WT $n = 10$; Elp3 cKO $n = 11$. 2 animals per group. **f** Number of axons: WT $n = 8$; Elp3 cKO $n = 9$. 2 animals per group. **h** 4 human primary fibroblast lines per group. Source data are provided with this paper.

regulating axonal transport in particular (Fig. 5i). Since these molecules are also detected in the cytosolic fraction together with Acss2 (S3, Fig. 2j–l), whose expression is reduced upon loss of Elongator (Fig. 2f, h), we cannot rule out the contribution of some cytosolic Acetyl-CoA generated by Acss2 in the vicinity of MTs, which would explain the remaining basal level of MT acetylation seen upon axonal transport blockade[30]. We observed comparable defects of α-tubulin acetylation and MT-dependent transport in fly, mouse, and human cells lacking Elongator activity, which, together with similar observations made in C. elegans[6], suggest a strong functional conservation of the EAA pathway across species.

A tight control of axonal transport is indeed very important to ensure the homeostatic activity of neurons by delivering cargos to distant regions, thereby controlling cytoskeleton maintenance as well as spreading long-range intracellular signaling that ensures cellular maintenance and function. Impairment of axonal transport is considered as an early pathological feature shared by several neurodegenerative disorders[2,48,49]. Along this line, our results obtained with FD fibroblasts support those previously described in neurons[28] and suggest that loss of Elongator activity contribute to neurodegeneration by interfering at least with MT-dependent transport in FD patients via impairment of ACLY expression. We did not observe differences in MT deacetylation activity in FD and WT fibroblasts (Supplementary Fig. 4h), suggesting that reduction of MT acetylation in FD fibroblasts does not arise from changes of HDAC6 expression, in contrast to what others have reported[28].

More generally, α-tubulin acetylation not only modulates axonal transport but also provides resistance to mechanical bending to MTs[50]. Therefore, by controlling the cytoskeleton integrity and function, the EAA pathway likely acts as a key regulator of neuronal fitness whose progressive dysfunction may contribute to neuronal aging and even underlie neurodegeneration. Thus, targeting this molecular pathway may open therapeutic perspectives to prevent the onset or the progression of neurodegenerative disorders like FD characterized by poor axonal transport and degeneration.

## Methods

**Mice.** All experiments performed in this study adhere to all relevant ethical regulations for animal testing and research. The animal work was approved by the ethical committee of the University of Liege under the license #18-2056.

Brains were harvested from mice at P0-P2 or E14.5. *FoxG1*:Cre $^{-/+}$/*Elp3* $^{lox/+}$ and *Elp3* $^{lox/lox}$ mice were time-mated to induce a conditional deletion of *Elp3* in the forebrain progenitors[33]. Brains were harvested from 10–16 months-old Elp1cKO and WT mice[27]. *Atat1* $^{+/-}$ mice were crossed together to obtain WT and KO mice[51]. Mice were housed under standard conditions (animals were bred in-house and maintained with ad libitum access to food and constant temperature (19–22 °C) and humidity (40–50%) under a 7am–7 pm light/dark cycle) and they were treated according to the guidelines of the Belgian Ministry of Agriculture in agreement with the European Community Laboratory Animal Care and Use

Regulations (86/609/CEE, Journal Official des Communautés Européennes L358, 18 December 1986). Neuronal cultures were prepared from dissected E14.5 mice brain cortices, followed by mechanical dissociation in HBBS (Sigma–Aldrich, H6648) supplemented with 1.5% glucose. Cells were cultured at confluence of ~70% (for 5 days) with Neurobasal Medium (Gibco, Invitrogen, 21103049) supplemented with 2% B27 (Gibco, Invitrogen, 17504044), 1% Pen/Strep (Gibco, Invitrogen, 15140122), and 1% Glutamax (Gibco, Invitrogen, 35050061) at 37 °C.

*Drosophila melanogaster.* Flies were kept at 25 °C in incubator with regular 12-h light and dark cycle. All crosses were performed at 25 °C, after two days hatched first instar larvae were transferred in a 29 °C incubator until use.

UAS-RNAi carrying lines were crossed with D42-Gal4-UAS:Syt1-GFP virgin females for axonal transport recording and immunohistochemistry, with D42-Gal4 for behavioral experiments, or with Elav-Gal4 for qPCR and WB analysis (See Table 1). All RNAi inserts sequences were validated by DNA sequencing.

**Human primary fibroblasts.** Fibroblasts were cultured on polystyrene culture flasks (Corning) at 37 °C with 5% $CO_2$ in DMEM (Gibco, Invitrogen, 11965092) medium supplemented with 10% Fetal Calf Serum (Gibco, Invitrogen, 10500056), 1 mM Sodium pyruvate (Gibco, Invitrogen, 11360070), 1 mM non-essential amino acids (Gibco, Invitrogen, 11140050). Primary fibroblasts from five FD patients and age-matched controls were purchased from Coriell biobank (www.coriell.org) (Table 2).

**Immunofluorescence.** E14.5 cortical neurons were plated on coverslips in 24 well plate at a density of 30,000 cells and cultured for 5DIV. Neurons were fixed by incubation in 4% PFA-sucrose in PBS for 20 min at RT and washed with PBS + 0.3% Triton-X. After washing, neurons were incubated in blocking solution (PBS + 0.3% Triton-X + 10% normal donkey serum) for 1 h at RT. Following overnight incubation with primary antibodies in blocking solution at 4 °C, washing, and incubation with secondary antibodies (PBS + 0.3% Triton-X + 1% normal donkey serum) at RT for 1 h and washing, coverslips were then mounted on microscope slide using Mowiol. Images were acquired using a Nikon A1Ti confocal microscope (60x oil lens) or Airyscan super-resolution module on a Zeiss LSM-880 confocal (63x oil lens).

Larvae were dissected in PBS to expose the brain and MNs. After dissection larvae were fixed with 4% PFA for 20 min at RT, washed with PBS + 0.2% (CSP staining) or 0.3% (α-Tubulin acetylation staining) Triton-X and incubated in blocking solution PBS, 0.2% (CSP staining), or 0.3% (α-Tubulin acetylation staining) Triton-X + 1% BSA for 30 min at RT. Following overnight incubation with primary antibodies at 4 °C, washing and incubation with secondary antibodies at RT for 2 h, the larvae were mounted on microscope slide in Mowiol. Images were acquired using a Nikon A1Ti confocal microscope (60x oil lens).

Primary fibroblasts from FD patients or controls were plated at the concentration of 5000 cells in a 96-well plate (uClear, Grinere). Cells were fixed with 4% PFA for 10 min, washed, permeabilized, and blocked for 1 h at RT with 5% fetal bovine serum in PBS + 0.1% Triton-X. Following overnight incubation with acetylated α-tubulin (Sigma-Aldrich) and β-tubulin (cell signaling) antibodies, cells were washed with PBS + 0.05% Triton-X times and incubated with secondary antibodies for 1 h at RT. Finally, the cells were washed, and remained in PBS for image acquisition. Images were acquired using a IN Cell Analyzer 2200 fluorescent microscope (General Electric) using a 60x air lens.

Fluorescence intensity levels were measured by Fiji (https://imagej.net/Fiji/Downloads). To analyze α-tubulin and acetylated α-tubulin levels of mice cortical neuron axons and fly MNs, ROIs of 30 μm long axonal segments accounting for their full width were used. For human primary fibroblasts, ROI of the entire cell were outlined. α-tubulin and acetylated α-tubulin levels were extracted from mean intensity levels. Background levels were subtracted and the ratio of acetylated α-tubulin/α-tubulin was calculated.

**Table 1 Drosophila melanogaster lines.**

| Line | Catalog number | Given name | Purchased from |
|------|----------------|------------|----------------|
| Elav-Gal4 | BDSC 458 | | Bloomington Drosophila Stock Center |
| huD42-Gal4 | BDSC 8816 | | Bloomington Drosophila Stock Center |
| UAS:Syt-GFP | BDSC 6925 | | Bloomington Drosophila Stock Center |
| UAS:RNAi Zpg | VDRC CG10125 | Control | Vienna Drosophila Resource Center |
| UAS:RNAi Elp1 | VDRC CG10535 | Elp1 KD | Vienna Drosophila Resource Center |
| UAS:RNAi Elp3 | VDRC CG15433 | Elp3 KD | Vienna Drosophila Resource Center |
| UAS:RNAi Atat1 | VDRC CG3967 | Atat1 KD | Vienna Drosophila Resource Center |
| UAS:RNAi Hdac6 | BDSC 51181 | Hdac6 KD | Bloomington Drosophila Stock Center |
| UAS:Human Elp3 | | hElp3 | Kindly provided by Patrick Verstreken |
| UAS:Human ACLY | VDRC CG8322 | hACLY | Vienna Drosophila Resource Center |

**Table 2 Human fibroblast lines.**

| Catalog number | Sex | Age | Sample description |
|----------------|-----|-----|--------------------|
| GM02036 | Female | 11 | Apparently Healthy |
| GM07492 | Male | 17 | Apparently Healthy |
| GM07522 | Female | 19 | Apparently Healthy |
| GM038 | Female | 3 | Apparently Healthy |
| GM05659 | Male | 14 | Apparently Healthy |
| GM02343 | Female | 24 | Familial Dysautonomia |
| GM04589 | Male | 16 | Familial Dysautonomia |
| GM04663 | Female | 2 | Familial Dysautonomia |
| GM04959 | Female | 10 | Familial Dysautonomia |
| GM04899 | Female | 12 | Familial Dysautonomia |

**Vesicular and mitochondrial transport recording in vitro and in vivo.** Axonal transport in mice cultured cortical neurons was recorded in microfluidics devices, prepared as described in[52]. Briefly, air bubbles were removed from mixed sylgard 184 elastomer (VossChemie Benelux, 1:15 ratio with curing agent) by centrifuging at $1000 \times g$ for 5 min. The liquid was poured into the microfluidic device mold and was cured by 3 h incubation at 70 °C incubator. Molds were cut and washed twice with 70% ethanol, air dried in biological hood and placed on 35 mm glass-bottom dishes (MatTek, P35G-0-20-C) previously coated with 0.1 mg/mL poly-D-lysine (overnight incubation at 4 °C prior to washes and air drying). To increase the adhesiveness, the microfluidic chambers and dishes were heated to 70 °C before assembling. E14.5 mouse cortical neurons were isolated and resuspended at $40000 \times 10^3$ cells/μL for seeding at a 70% confluency into the microfluidic somal channel. After plating, the axonal and somal compartments of the microfluidic devices were filled with culture medium supplemented with 50 ng/ml or 20 ng/ml BDNF (PeproTech, 450-02), respectively. Labeling were done on after 5DIV by adding 1 μM LysoTracker® Red DND-99 (ThermoFisher Scientifics, L7528), MitoTracker® Green FM (ThermoFisher Scientifics, M7514) or MitoTracker® Deep Red FM (ThermoFisher Scientifics, M22426) 30 min prior to time-lapse recordings. Axonal transport recording in Drosophila Melanogaster MNs were done on 3rd instar larvae expressing UAS:RNAi and Syt1-GFP. Larvae were anaesthetized with ether vapors (8 min) and mounted dorsally on microscope slide using 80% glycerol. Recordings of mice cortical neurons and Drosophila melanogaster MNs were performed on an inverted confocal microscope Nikon A1Ti (60x oil lens) or Zeiss LSM-880 confocal (63x oil lens) at 600 ms frame interval.

Intracellular transport in human primary fibroblasts was recorded 3 days after plating, by adding 1 μM LysoTracker® Red DND-99 (ThermoFisher Scientifics, L7528) 30 min prior to time-lapse recordings. Recordings of human primary fibroblasts were performed using in-cell 2200 (General Electric) using 60X air lens, at 2 s frame intervals, using temperature (37 °C) and $CO_2$ control.

Video analysis was performed by generation of kymographs for single-blind analysis using ImageJ plugin-KymoToolBox (fabrice.cordelieres@curie.u-psud.fr). Average velocity is defined as the cumulative distance traveled by an organelle divided by the total amount of time, therefore it includes the pausing time. Moving velocity is defined by the cumulative distance traveled anterograde or retrograde divided by the time traveled anterograde or retrograde, therefore it excludes the pausing time. Moving velocity for anterograde is represented by positive values and retrograde by negative values. Pausing time is defined as the percentage of time that an organelle spends at a velocity lowered than 0.1 μm/sec out of the total traveling time. All kymographs show anterograde (left to right direction) and retrograde (right to left) moving vesicles. For analysis of axonal transport in Drosophila melanogaster, the StackReg plugin was used to align frames.

**Plasmids and drug treatments.** For silencing of Elp3 in mouse cortical neurons we inserted sh-Elp3 5'GCACAAGGCUGGAGAUCGGUU3' or a control sequence sh-Control 5'-TACGCGCATAAGATTAGGG-3' previously described in[25,53]. The viral packaging vector is PSPAX2 and the envelope is VSV-G. The lentiviral vector is pCDH-cmv-EF1-copGFP (CD511B1), and the promotor was replaced by a U6 promoter. For expression of human ACLY in E14.5 mice cortical neurons, cells in suspension were transfected with pEF6-Acly (Addgene plasmid # 70765, a gift from Kathryn Wellen)[54] or pEF6 (control) and GFP (from VPG-1001, Lonza) using Mouse Neuron Nucleofector® Kit (VPG-1001, Lonza) according to the manufacturer's protocol. GFP-positive neurons were used for analysis. To assay the subcellular localization of Acly in respect to lamp1-positive late endosomes/lysosomes in mouse cortical neurons, pCAG mEmerald-LAMP1 provided by F. Polleux (Columbia University, New York, USA) was transfected using Mouse Neuron Nucleofector® Kit (VPG-1001, Lonza) according to manufacturer's protocol. For expression of human ACLY in cultured human primary fibroblasts, cells were transfected with pEF6-Acly (Addgene plasmid # 70765, a gift from Kathryn Wellen)[54] or pEF6 (control) and GFP (from VPG-1001, Lonza) using Lipofectamine® 2000 according to manufacturer's protocol. GFP-positive cells were used for analysis. Tubastatin A (TBA, 20 μM) or Hydroxy-citrate (HCA, 3 mM or 10 mM) were dissolved in Ultra-Pure Water (UPW) and added to cell cultures 2 h or 8 h, respectively, prior to recording.

**Real-time quantitative PCR analysis (qRT-PCR).** One mouse cortex or ten adult fly heads were collected in TRIzol Reagent (Ambion, Life Technologies) followed by RNA extraction performed following the manufacturer's instructions. After DNAse treatment (Roche), 1 μg of RNA was reverse transcribed with RevertAid Reverse Transcriptase (Fermentas). RT-qPCR was performed using Quant Studio (Thermo) and TaqMan primers for mice, or a Light Cycler 480 (Roche) with Syber Green mix for Drosophila melanogaster. Analyses were done using the $2^{-\Delta\Delta CT}$ method[55] with the primers listed in Table 3.

**Protein aggregation assay.** Cultured mice cortical neurons or human primary fibroblasts at 60% confluency were incubated with PROTEOSTAT® (Enzo Life-Sciences) to detect protein aggregates.

**Locomotion activity and climbing assays.** Larval crawling speed assays were performed by placing 3rd instar larvae in the center of 15-cm petri dishes coated with 3% agar, as previously published[56]. Velocities were extrapolated by measuring the distance traveled in 1 min. The climbing assay was performed as previously described in[57], by measuring the average ratio of successful climbs over 15 cm for 10 adult flies.

**Western blot.** Mouse brain cortices, adult fly heads, or human primary fibroblasts were quickly homogenized on ice in RIPA buffer, or 320 mM sucrose, 4 mM HEPES buffer for subcellular fractions. Protease inhibitor cocktail (Roche, P8340 or Sigma-Aldrich, S8820) and 5 μM Trichostatin-A (Sigma-Aldrich, T8552) were added to inhibit protein degradation and α-tubulin deacetylation.

Immunoblotting was performed with the primary and secondary antibodies listed in Supplementary Table 1. We used 2 μg of protein lysate for α-tubulin acetylation analysis and 20–30 μg for all other proteins. Nitrocellulose membranes were imaged using Amersham Imager 600 (General Electric, 29083461) and band densitometry was measured using FIJI.

**Subcellular fractionation.** Subcellular fractionation of frozen mice brain cortex was performed as previously described[30]. Tissues or cells were homogenized in 320 mM sucrose, 4 mM HEPES, pH 7.4 (fractionation buffer). The homogenate was centrifuged for 10 min at $1000 \times g$ to obtain a pellet (P1) (nuclear fraction) and a post-nuclear fraction (S1). The supernatant (S1) was centrifuged for 40 min at $12,000 \times g$ to obtain large membranes (P2) and vesicle-enriched cytosolic fraction (S2). Finally, S2 was centrifuged again for 90 min at $120,000 \times g$ using a TLA 120.1 fixed angle rotor in Optima TLX Benchtop Ultracentrifuge to obtain vesicle

**Table 3 List of PCR primers.**

| Gene | Forward | Reverse | Organism |
|------|---------|---------|----------|
| *Acly* | Hs00982738_m1 (Thermo) | | mice |
| *Gapdh* | Mm99999915_g1 (Thermo) | | mice |
| TBP | CCACGGTGAATCTGTGCT | GGAGTCGTCCTCGCTCTT | fly |
| *Elp1* | TCGGCGGTTCCTTTCCAAAC | GGTCCGATGCAACTTCAGATT | fly |
| *Elp3* | AAGAAGTTGGGCGTGGGATT | ATCCTTTTTGGCTTCGTGCG | fly |

fraction (P3) and a cytosolic fraction (S3). The vesicle fraction was gently washed once with fractionation buffer, and resuspended by vigorous pipetting with the same buffer.

**Mass spectrometry analysis.** Pellets from three independent samples of pooled vesicles isolated from the brain cortex of WT or *Elp1 KD* mice were solubilized using 5% SDS. The samples were dissolved in 10 mM DTT 100 mM Tris and 5% SDS, sonicated, and boiled at 95 °C for 5 min. The samples were precipitated in 80% acetone. The protein pellets were dissolved in 9 M Urea and 100 mM ammonium bicarbonate than reduced with 3 mM DTT (60 °C for 30 min), modified with 10 mM iodoacetamide in 100 mM ammonium bicarbonate (room temperature for 30 min in the dark), and digested in 2 M Urea, 25 mM ammonium bicarbonate with modified trypsin (Promega), overnight at 37 °C in a 1:50 (M/M) enzyme-to-substrate ratio. The resulting tryptic peptides were desalted using C18 tips (Harvard) dried and resuspended in 0.1% Formic acid. Samples were analyzed by LC-MS/MS using Q Exactive plus mass spectrometer (Thermo) fitted with a capillary HPLC (easy nLC 1000, Thermo). The peptides were loaded onto a homemade capillary column (20 cm, 75 micron ID) packed with Reprosil C18-Aqua (Dr Maisch GmbH, Germany) in solvent A (0.1% formic acid in water). The peptides mixture was resolved with a (5 to 28%) linear gradient of solvent B (95% acetonitrile with 0.1% formic acid) for 60 min followed by a gradient of 28–95% for 15 min, and 15 min at 95% acetonitrile with 0.1% formic acid in water at flow rates of 0.15 μl/min. Mass spectrometry was performed in a positive mode using repetitively full MS scan followed by high collision-induced dissociation (HCD, at 35 normalized collision energy) of the 10 most dominant ions (>1 charges) selected from the first MS scan. The mass spectrometry data were analyzed using the MaxQuant software 1.5.1.2. (//www.maxquant.org) using the Andromeda search engine, searching against the mouse uniprot database with a mass tolerance of 20 ppm for the precursor masses and 20 ppm for the fragment ions. Peptide- and protein-level false discovery rates (FDRs) were filtered to 1% using the target-decoy strategy. Protein table were filtered to eliminate the identifications from the reverse database, and common contaminants and single peptide identifications. The data were quantified by label-free analysis using the same software, based on extracted ion currents (XICs) of peptides enabling quantitation from each LC/MS run for each peptide identified in any of the experiments. The mass spectrometry proteomics data have been deposited to the ProteomeXchange Consortium via the PRIDE partner repository with the dataset identifier PXD021186.

**In vitro α-tubulin assay.** In vitro α-tubulin acetylation assay was performed as previously described[30]. Briefly, 96-well half-area plates (Greiner, 674061) were coated with 5 μg tubulin from HeLa cells or bovine brain (Cytoskeleton, catalog no. HTS02-A) in 25 μL of ultrapure water for 2.5 h at 37 °C, blocked (PBS + 3% BSA + 3% skim milk + 3% FBS) for 1 h at 37 °C, and washed with PBS + 0.05% Tween 20. Wells were incubated for 2 h at 37 °C with shaking at 140x *g* with 25 μg of the total, S2, S3, or P3 fractions isolated from P0 mouse brain cortex previously diluted in 2× histone acetyltransferase (HAT) buffer (Sigma–Aldrich, EPI001A) supplemented with protease inhibitor cocktail, 5 μM trichostatin, and 50 μM acetyl-CoA (Sigma-Aldrich, A2056) or vehicle (H20) for control. After washes and overnight incubation at 4 °C with acetylated a-tubulin antibody (1:2000) in blocking buffer (PBS + 0.05% Tween 20 + 3% BSA), wells were washed and incubated for 2 h at 37 °C with peroxidase-conjugated goat antimouse antibody (1:5000) in antibody blocking buffer. Wells were then washed and incubated with trimethylboron/E (Merck Millipore, ES001) where reaction was thereafter stopped with H2SO4. Following the manufacturer's instructions, the HAT activity colorimetric assay kit (Sigma–Aldrich, EPI001) was used to measure CoA release.

**α-Tubulin deacetylation assay.** α-Tubulin deacetylation assay was performed as previously described in[38]. Briefly, 96-well half-area plates (Greiner, 674061) were coated with 1 μg tubulin in 50 μl of ultra-pure water for 2.5 h at 37 °C, followed by blocking (PBS, 3% BSA, 3% skim milk, 3% fetal bovine serum) for 1 h at 37 °C, and washing with PBS + 0.05% Tween-20. 10 μg of cytosolic fraction isolated from newborn mice brain cortex in deacetylation buffer (50 mM Tris-HCl at pH 7.6, 120 mM NaCl, 0.5 mM EDTA) supplemented with protease inhibitor cocktail and phosphatase inhibitor (to avoid dephosphorylation of HDAC6) were added per well for incubation of 4 h at 37 °C with shaking at 120x *g*. The wells were washed and incubated overnight at 4 °C with acetylated α-tubulin antibody (1:2000) in blocking buffer (PBS + 0.05% Tween-20 + 3%BSA). Wells were subsequently

washed and incubated for 2 h at 37 °C with Peroxidase-conjugated Goat antimouse antibody (1: 5000) in antibody blocking buffer. After the final washes, the peroxidase activity was probed with TMB/E (ES001, Merck Millipore), and the reaction stopped with H₂SO₄.

**Acly activity assay.** Acly activity assay was measured as previously described in[41]. Briefly, cells or brain extracts were disrupted by passing them 15 times in 25-gauge needle in 100 mM Trish-HCL buffer, supplemented with protease and phosphatase inhibitors. 5ug of cells/brain lysate were added to the reaction mix (200 mM Tris-HCL pH 8.4, 20 mM MgCl₂, 20 mM sodium citrate, 1 mM DTT, 0.1 mM NADH (Sigma-Aldrich, N8129), 6 U/mL Malate dehydrogenase (Sigma-Aldrich, M1567), 0.5 mM CoA (Sigma-Aldrich, C3019) with or without ATP (Sigma-Aldrich, A1852). Acly activity was measured every 100 s for 4 h, in the volume of 50ul in a 384 well plate using 340 nm OD read. Acly specific activity was calculated as the change in absorbance in the presence or absence of ATP. For statistical comparison Acly activity was defined as the slope from the linear range of the reaction.

**Acetyl-CoA sample preparation and LC-MS/MS analysis.** Acetyl-CoA was extracted as previously described[58]. Briefly, cold methanol (500 μl; −20 °C) was added to the cell pellets, and the mixture was shaken for 30 s (10 °C, 2000 r.p.m., Thermomixer C, Eppendorf). Cold chloroform (500 μl; −20 °C) was added, the mixture was shaken for 30 s, and 200 μl of water (4 °C) added afterwards. After 30 s shaking and 10 min on ice, the mixture was centrifuged (21,000 × g, 4 °C, 10 min). The upper layer was collected and evaporated. The dry residue was re-dissolved in eluent buffer (500 μl) and centrifuged (21,000 × g, 4 °C, 10 min) before placing in LC-MS vials. Acetyl CoA was analyzed as previously described[59]. Briefly, the LC-MS/MS instrument consisted of an Acquity I-class UPLC system and Xevo TQ-S triple quadrupole mass spectrometer (both Waters) equipped with an electrospray ion source. LC was performed using a 100 × 2.1-mm i.d., 1.7-μm UPLC Kinetex XB-C18 column (Phenomenex) with mobile phases A (10 mM ammonium acetate and 5 mM ammonium hydrocarbonate buffer, pH 7.0, adjusted with 10% acetic acid) and B (acetonitrile) at a flow rate of 0.3 mL min-1 and column temperature of 25 °C. The gradient was set as follows: 0–5.5 min, linear increase 0–25% B, then 5.5–6.0 min, linear increase till 100% B, 6.0–7.0 min, hold at 100% B, 7.0–7.5 min, back to 0% B, and equilibration at 0% B for 2.5 min. Samples kept at 4 °C were automatically injected in a volume of 5 μl. Mass spectrometry was performed in positive ion mode, monitoring the MS/MS transitions m/z 810.02 → 428.04 and 810.02 → 303.13 for acetyl-CoA. Spikes of defined amounts of AcCoA were added to the samples to confirm the absence of signal inhibition (matrix effect) in the analyzed extracts. Quantification of AcCoA was done against an external calibration curve with 1–1000 ng mL−1 range of AcCoA concentrations using TargetLynx software (Waters).

**Polysomal fractionation.** Polysome-bound RNA was purified from the cortical extract of P0 pups according to Laguesse et al.[45]. Briefly, fresh mouse cortex was snap-frozen in 1.5 mL Eppendorf tube, pulverized in liquid nitrogen with a pestle, and kept on dry ice for 5 min. The powder was resuspended in 1 mL lysis buffer (10 mM Trish pH 8.0, 150 mM NaCl, 5 mM MgCl₂, 1% NP-40, 0.5% sodium deoxycholate, 40 mM dithiothreitol, 10 mM Ribonucleoside Vanadyl Complex, 200 μg/mL cycloheximide and 400 U/mL RNAsin). 200 μL of the homogenate was used to isolate total RNA using TRIzol reagent. The remaining 800 μL were centrifuged for 10 s at 12.000 × g to discard nuclei, and the supernatant was collected. Ribosomes were further released by adding 800 μL of 2X extraction buffer (200 mM Tris pH 7.5, 300 mM NaCl, and 200 μg/mL cycloheximide) and samples were centrifuged at 12,000 × g for 5 min at 4 °C to remove mitochondria and membranous debris. The resulting supernatant was loaded onto a 8 mL 15–45% sucrose gradient and centrifuged at 100,000 × g (at r_min), 4 °C for 2 h without brake, by using a SW4Ti rotor (Beckman Coulter). The four first 500 μl bottom fractions were collected and digested with proteinase K buffer (400 μg/mL Proteinase K, 10 mM EDTA, 1% SDS) at 37 °C for 30 min, to be thereafter extracted with phenol-chloroform. Polysomal fractions were assayed for purity by assessing the presence of 28 S and 18 S ribosomal RNA bands on an agarose gel, and by measuring absorbance at 254 nm, as previously described[60].

**Puro-PLA and immunostaining.** Puro-PLA was performed as previously described[61] to label newly synthesized proteins of interest. In brief, neurons cultured 5 days in vitro were incubated with 10 μg/mL puromycin for 4 min, quickly

washed with warm PBS, and fixed for 10 min in 4% PFA-sucrose. After fixation and permeabilization, the Duolink assay (DUO92007; DUO92004; DUO92002 Sigma) was performed as recommended by the manufacturer using the following primary antibodies: puromycin 1:75 (PMY-2A4 DSHB); Acly 1:100 (OAGA04026 Aviva Systems Biology). Following puro-PLA, cells were post-fixed for 10 min in PFA-sucrose, washed with PBS, blocked with 5% donkey serum in PBS, and labelled with antitubulin antibody 1:1000 (ab6160) in Duolink Antibody Diluent (Sigma) for 1 h. After incubation with fluorophore-labelled secondary antibody for 1 h and further washes, cells were mounted with Fluorescence Mounting medium (S3023 Agilent Dako). Images were acquired with a Zeiss LSM-880 microscope using × 40 oil objective. Images were processed with ImageJ (NIH).

**Cycloheximide pulse-chase assay.** Primary fibroblasts from FD patients or healthy controls were cultured in 6-well plates at 80% confluency before treatment with 50 µg/mL cycloheximide (CHX) (C7698 Sigma) for 0 h, 4 h, 6 h, 8 h. All samples were harvested simultaneously by washing the cells in each well with ice-cold PBS, adding in each well 150 µL cell lysis buffer (50 mM Tris-HCl, pH 8, containing 150 mM NaCl, 1% NP-40 supplemented with complete proteinase inhibitor cocktail (Roche)) and subsequent mechanical scraping. Following incubation on ice for 30 min, samples were centrifuged for 20 min at 13,000 x g to collect the supernatant. Protein concentration of each sample was measured by using the Protein Assay Kit (Pierce Biotechnology, Rockford) according to the manufacturer's instructions. All samples were resuspended in SDS buffer before WB analysis.

**Co-immunoprecipitation analysis.** Co-immunoprecipitation experiments were performed in HEK293 cells (Merck Chemicals, Brussels). Pierce Crosslink immunoprecipitation kit (#26147) was used following the manufacturer's instructions. For Elp3-FLAG and Myc-Acly analysis 4 µg of Myc (Cell signaling), Flag (Sigma-Aldrich), or general mouse IgG (Millipore) were crosslinked using DSS to protein A/G. For detection by WB of Elp3-Flag or Myc-Acly Elp3 (Jesper Svejstrup) antibody and Acly (cell signaling) were used.

**Statistics and Reproducibility.** Every experiment was repeated independently at least two times under single-blinded condition and statistical analyses were generated with GraphPad Prism Software 7.0. For statistical analysis, data normality was assessed via D'Agostino-Pearson omnibus normality test or Shapiro–Wilk normality test. Comparison between two groups with normally distributed samples was performed with two-sided unpaired $t$ test, while comparisons among more than two groups with one-way or two-way analysis of variance (ANOVA) followed by Sidak's, Dunn's or Dunnet's post hoc test. If normality was not reached, Mann-Whitney or Kruskal–Wallis test were performed, to compare between two groups or among more than two groups, respectively.

**Reporting summary.** Further information on research design is available in the Nature Research Reporting Summary linked to this article.

## Data availability

The proteomic data generated in this study have been deposited in the ProteomeXChange database and can be accessed via the CX accession number: PXD021186. Proteomic data supporting the findings of this study are available within the paper and its supplementary information files. Source data are provided with this paper.

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

## Acknowledgements

We thank Maria M. Magiera and Carsten Janke for providing non acetylated Hela Tubulin. We thank Patrik Verstreken for sharing Elp3KD flies, M. Nachury for sharing ATAT1 antibody and X.J. Yang for providing the Atat1 KO mice, T. Lahusen from American Gene Technologies for creating viral sh-RNA particles, the Smoler Proteomics Center at the Technion for proteomic analysis and E. Even for graphical design. All graphical figures were created with BioRender.com. We are grateful to Francesca Bartolini and Marina Mikhaylova for their constructive feedback on the manuscript as well as to all members of the Nguyen and Weil laboratories for their critical reading. L.N. is Director from F.R.S- F.N.R.S. This work was supported by the F.R.S.-F.N.R.S. (Synet; EOS 0019118F-RG36), the Fonds Leon Fredericq (L.N.), the Fondation Médicale Reine Elisabeth (L.N.), the Fondation Simone et Pierre Clerdent (L.N), the Belgian Science Policy (IAP-VII network P7/20 (L.N.)), and the ERANET Neuron STEM-MCD and NeuroTalk (L.N.); grants from Agence Nationale de la Recherche (ANR-18-CE16-0009-01 AXYON (F.S.); ANR-15-IDEX-02 NeuroCoG (F.S.) in the framework of the "Investissements d'avenir" program); Fondation pour la Recherche Médicale (FRM, DEI20151234418, F.S.). A.E.'s stay at GIGA Research Institute of the University of Liège was funded by EMBO Short-Term Fellowships (ASTF 174-2016), A.E., M.S., and M.W.'s research was supported by the Israel Science Foundation (grant no. 1688/16). The authors declare no competing financial interests.

## Author contributions

A.E., G.M., M.W., S.T. and L.N. designed the study. A.E., G.M. and S.T. performed and interpreted most experiments. M.S. and R.L.B. are to be considered as equal contributors. R.L.B. contributed to the *Drosophila* work with help of N.K.. M.S. contributed to cell culture and molecular work with L.B., Ar.B., S.I. and S.L. Al.B. performed the LC-MS/MS analysis. P.D. and I.D. maintained *Elp1 KD* mice colonies and provided brain material. F.S., A.C., B.B and J.M.R provided guidance and help for experiments. M.W. and L.N. contributed to data interpretation; and A.E., G.M., S.T., M.W. and L.N. wrote the manuscript with input from all coauthors.

## Competing interests

The authors declare no competing interests.
