## [Peer Review File · Nature Communications]

REVIEWER COMMENTS

Reviewer #1 (Remarks to the Author):

In this article, Even and colleagues investigate the mechanism by which Elongator modulates MT acetylation and transport. They demonstrate that Elongator controls the protein levels of the ATP citrate lyase ACLY, an enzyme that produces acetyl-CoA. This molecule provides the acetyl groups required for MT acetylation by ATAT1. Accordingly, the authors detect no additive effect between Elongator and ATAT1 loss of function on MT acetylation and MT-based transport, and observe similar defects upon ACLY loss of function, arguing for a role in a common pathway. They demonstrate a conservation of these functions between mouse and flies, and show that ACLY overexpression rescues Elongator loss of function defects. Finally, they demonstrate that Elongator-mutated cells from FD patients display reduced ACLY levels, and that ACLY overexpression rescues MT acetylation and transport defects in these cells. The main findings are novel and generally well-supported by the data. I however have several comments that need to be addressed.

Major comments

1/ The results of figure 2, while convincing, are not presented in the most logical way, which can be misleading. First, the authors test the activity of ATAT1 in Elp3 cKO extracts using an in vitro assay where exogenous acetyl-CoA is added. They state that, "In strike contrast to brain extracts from Atat1 KO mouse pups, we did not observe any differences in MT acetylation level between cortical extracts from Elp3 cKO and WT mice". They conclude that "loss of Elongator activity does not impair the acetylation of α -tubulin by changing the expression or activity of Hdac6 or Atat1. This conclusion is valid but this experiment makes a much more important point. It shows that, unlike what is seen on the levels of MT acetylation by IF or WB in Elp3 cKO, no effect is observed in the presence of exogenously added Acetyl-CoA, pointing to a miss-regulation of the production of this metabolite.

It is also not clear why the authors next investigate specifically the vesiculome. They "postulate that Elongator could interfere with MT acetylation and axonal transport via reduction of AclY expression", which is perfectly reasonable. But why not directly present the proteome? Especially because they next demonstrate reduced ACLY expression in Elp3 cKO embryos. The vesicular aspect of the story, while convincing and interesting, should come next, to show that ACLY is enriched on vesicles, and is therefore likely to act from there. The current organizing gives the false impression that the defect is specifically occurring on vesicles.

Related to this, it was not clear how the authors identified ACLY specifically? Was it one of the top hits of the vesiculome, or was this an educated guess? Either is fine, but it should be explained. Also, what is the proportion of proteins whose levels are affected in Elp3 cKO (see also next comment)?

2/ A major finding of this study is the role of the Elongator complex in the regulation of ACLY protein levels, thereby identifying the cause of MT acetylation defects in Elp3 KO cells and FD patient cells. While the authors argue that the identification of the mechanism for this is out of the scope of this study, they rule out an effect of Elongator on ACLY translation. To support this, they state that ACLY expression in mouse and fly Elongator models rescues the acetylation defects. This argument is however flawed, as defective translation could very well be rescued by strong overexpression of the ACLY transgene. What are the levels of human ACLY expression, as compared to endogenous ACLY? Ideally, an experiment disproving translation defects as a cause of the phenotype should be performed. This is particularly critical as protein translation is a major function of the Elongator complex. As stated in the previous comment, it would be important to estimate the fraction of proteins for which levels are defective.

3/ A precise description of how velocity quantifications have been performed is missing (in mat & met section). The reviewer understands that these graph representations may be classical, but the authors should clarify in the legend (or at least in the methods) what they mean by average velocity and moving velocity, and how this is measured. Also, for moving velocity representations, the authors should indicate to which direction correspond positive and negative velocities (I assume anterograde and retrograde but it should be indicated). Related to this, they should also explain why average velocities are represented in absolute values, while moving velocities are represented in positive and negative values. Finally, it is not clear how a pausing time can be expressed in percentage. Again, if this is standard, it should be explained in the legend. By percentage of pausing time, do they mean percentage of time spent in pause?

4/ While the quantifications are clear, the corresponding kymographs are often not very representative (Figures S1g, S3c, S3g...). The authors should replace some of them with examples that represent the quantifications more accurately. If the authors consider these as representative, they should highlight the events that are differently affected. Finally, and related to the previous comment, there are many large immobile structures. Are they all included in the quantification and how?

Specific comments

1/ The authors should quantify ACLY enrichment on the P3 fraction (Fig 2k).

2/ Figure S4h: The Proteostat signal appears brighter in the FD cells compared to control. Is the image not representative, or is there an effect on the size / number of aggregates per cell?

Minor comments

line 30: (MT)-based transport is an evolutionary conserved process

line 33: the mechanisms that control

line 34: remain

Line 249: "In extracts fibroblasts". Rather in fibroblast extracts

286: "is currently under investigation in the laboratory". The authors should avoid such statements.

Figure 1a: is this experiment performed in a microfluidic device? The device is only described in 1b but the legend states 1a is a microfluidic experiment too.

Figure 2i: Absorbance

Figure 2k: Although this is most likely to be due to a technical problem (during protein transfer for instance), the fact that there is almost no signal for Atat1 in the S1 fraction is incoherent. Ideally, another WB should be provided.

Figure 3a: The image for reduced acetylated Tubulin in Acly KD is not very convincing (especially when compared to 3f, which is clear). (Also, the scale bar is mispositioned.)

Figure 4C: A very bright structure (both for Tub and ac-Tub) is shown in FD + Acly cells. What is it? If often observed, could it bias signal measurements?

Reviewer #2 (Remarks to the Author):

Summary:

In this manuscript, the authors document a role for ATP-citrate lyase (ACLY) in the regulation of Ataxin-1-dependent α -tubulin acetylation and axonal transport. Previous work identified a role for the Elongator complex in regulating microtubule acetylation, though mechanisms remained to be defined. Overall, the manuscript provides convincing evidence in support of their core claims that ACLY is present in vesicles along with Ataxin-1, that its levels are reduced in the absence of Elongator, and that it regulates α -tubulin acetylation downstream of Elongator. Both murine and fly models are used. Very interestingly, relevance of this pathway in Friedreich's ataxia (FA) patient fibroblasts is shown.

Major point:

1. Authors do a good job of documenting a key role for ACLY in the regulation of α -tubulin acetylation and axonal transport downstream of Elongator. However an important notion put forward here is that it is localized acetyl-CoA production by vesicular ACLY that is crucial for ATAXIN-1-dependent tubulin acetylation. This is an important novel aspect of this paper, and in my view should be examined more critically.

o ACLY and ACSS2 are the two major sources of cytosolic acetyl-CoA and ACSS2 can compensate for ACLY deficiency in at least some contexts. Why is ACLY critical for this function- is ACSS2 expressed? Is it present in vesicles? Does acetate supplementation rescue phenotypes in the absence of Elongator or ACLY? Does ACLY physically interact with ATAXIN-1? Mews et al (Nature 2017) reported a key role for ACSS2 in regulating histone acetylation in neurons, and found it to be predominantly nuclear. Is it possible that the nuclear localization of ACSS2 precludes its involvement in α -tubulin acetylation regulation? Investigating these ideas would help to establish whether localized acetyl-CoA production is really a core part of this pathway.

o This may not be feasible at this time, but the model could really be nailed down if ACLY could be localized away from vesicles, or if conditions that regulate ACLY vesicular localization could be identified, and loss of MT acetylation observed.

Minor:

1. I urge the authors to consider changing the title. "Fuels" implies a direct metabolic input into axonal transport, while the mechanism proposed is a post-translational modification. The language used in the running title seems more appropriate.

2. Please clarify in figures or legends the conditions being compared in each statistical analysis.
3. Fig. 1K: Elp3 KD; ATAT1 KD appear to have additive effects. Please also include statistics comparing this to Elp3 KD alone.
4. Perhaps I missed it, but please blot for ATAT1 comparing WT to ATAT1 KO. There is sometimes a double band and sometimes not in the blots shown for ATAT1, and it would be helpful to clarify whether one or both of the bands are ATAT1.
5. Fig 3i: absorbance is misspelled in y axis

Reviewer #3 (Remarks to the Author):

The manuscript by Even et al. claims to identify ATP-citrate lyase (Acy) as a rate-limiting enzyme that modulates activity of Atat1, which acetylates alpha-tubulin. They show that neurons lacking Elongator complex have decreased Acy levels which contributes to deficits in tubulin acetylation and MT-dependent transport in projection neurons. They further report similar deficits in MT-dependent transport in fibroblasts from Familial Dysautonomia (FD) patients carrying mutations in Elongator complex. While this previously unknown role of Acy in the function of Elongator complex acetylating tubulin is an important finding and could be of general interest to the cytoskeleton field, I have several major concerns for the manuscript as it stands now, outlined below:

1. The authors perform bulk of their most important measurement of the ratio of acetylated α -tubulin to total tubulin via immunofluorescence (IF) images. Since they do have a good antibody for acetylated α -tubulin (as shown in supplementary Figure S1) to detect in western blots, and given that extracting protein measurements using IF images can be tricky and susceptible to huge variations across samples owing to different staining levels, it is important that the authors complement the IF data with quantitation using western blots.
2. For some of the axonal transport data shown in main Figures, the authors never provide any raw data in the form of kymographs, as in Figures 1c,d. It is imperative that the authors provide raw data for all of their quantification in the main figures. Furthermore, in the kymographs shown in the supplemental figures – kymographs for some of the rescue experiments do not really reflect the phenotype being reverted back to WT. For example,
HDAC6 KD rescue kymographs in Figure S1q
Elp3 KD, Acy rescue kymograph in Figure S3c

Acly rescue kymographs in Figure S3g

None of these kymographs represent the supporting quantitation and the claims the authors make in the text. Are these just a one-off? If yes, can the authors pick more representative images. Or are there concerns about the phenotypes not being robust enough?

3. I have several comments on Figure 2.

(i) I am surprised with the result in Figure 2d where the authors report tubulin acetylation level of Elp3 cKO cortical extracts comparable to WT extracts. If Elongator complex is required for tubulin acetylation, how do you explain this result?

(ii) Does lack of Elp3 affect tubulin de-acetylation as well as shown in Figure 2e? The authors should perform this experiment with Atat1 KO as well, as done in Figure 2d

(iii) In addition to the data in Figures 2m, n the authors should also show Acly levels in the whole extract of WT vs Elp3 cKO to determine if the overall levels of Acly are also affected?

4. How do the authors explain rescue of axonal deficits in Elp3 KO neurons with HDAC6 KD? This is an interesting result and is worth some discussion in the paper – if the authors think axonal transport is determined by a fine balance of acetylated vs de-acetylated tubulin? If this model is in fact true, can HDAC6 KD revert the axonal deficits they report in Acly KD neurons shown in Figure 3?

5. The authors constantly keep switching between in vivo imaging of Drosophila neurons and neuronal cultures from various mouse lines. While the results being consistent across species is encouraging, the differences in experiments performed across these two model systems sometimes makes the results harder to understand/interpret. For example, in Figure 1 the authors start out with neurons from Elp3 cKO mice and then switch to experiments with neurons from Atat1 KO mice harboring RNAi for Elp3. They perform these experiments in the fly model as well and rescue the phenotype with KD of Hdac6, which is not done with neurons from Elp3 cKO mice. The authors finish up the data set with Elp3 cKO mice again in supplementary figure S1. This seems to be the case across multiple figures throughout the paper. Can the authors consolidate the data across figures that helps with data interpretation and will also make it easier for the reader.

Response to referees

Reviewer #1

In this article, Even and colleagues investigate the mechanism by which Elongator modulates MT acetylation and transport. They demonstrate that Elongator controls the protein levels of the ATP citrate lysase ACLY, an enzyme that produces acetyl-CoA. This molecule provides the acetyl groups required for MT acetylation by ATAT1. Accordingly, the authors detect no additive effect between Elongator and ATAT1 loss of function on MT acetylation and MT-based transport, and observe similar defects upon ACLY loss of function, arguing for a role in a common pathway. They demonstrate a conservation of these functions between mouse and flies, and show that ACLY overexpression rescues Elongator loss of function defects. Finally, they demonstrate that Elongator-mutated cells from FD patients display reduced ACLY levels, and that ACLY overexpression rescues MT acetylation and transport defects in these cells. The main findings are novel and generally well-supported by the data. I however have several comments that need to be addressed.

We thank the reviewer for the enthusiasm about our data and for providing a constructive feedback.

Major comments

1/ The results of figure 2, while convincing, are not presented in the most logical way, which can be misleading. First, the authors test the activity of ATAT1 in Elp3 cKO extracts using an in vitro assay where exogenous acetyl-CoA is added. They state that, “In strike contrast to brain extracts from Atat1 KO mouse pups, we did not observe any differences in MT acetylation level between cortical extracts from Elp3 cKO and WT mice”. They conclude that “loss of Elongator activity does not impair the acetylation of α -tubulin by changing the expression or activity of Hdac6 or Atat1. This conclusion is valid but this experiment makes a much more important point. It shows that, unlike what is seen on the levels of MT acetylation by IF or WB in Elp3 cKO, no effect is observed in the presence of exogenously added Acetyl-CoA, pointing to a miss-regulation of the production of this metabolite. It is also not clear why the authors next investigate specifically the vesiculome. They “postulate that Elongator could interfere with MT acetylation and axonal transport via reduction of Acly expression”, which is perfectly reasonable. But why not directly present the proteome? Especially because they next demonstrate reduced ACLY expression in Elp3 cKO embryos. The vesicular aspect of the story, while convincing and interesting, should come next, to show that ACLY is enriched on vesicles, and is therefore likely to act from there. The current organizing gives the false impression that the defect is specifically occurring on vesicles. Related to this, it was not clear how the authors identified ACLY specifically? Was it one of the top hits of the vesiculome, or was this an educated guess? Either is fine, but it should be explained. Also, what is the proportion of proteins whose levels are affected in Elp3 cKO (see also next comment)?

We thank the reviewer for providing such eye opening remarks. We fully agree that the initial figure 2 was not well organized and was lacking rational to justify the focus on Acly and on vesicles. As mentioned by the reviewer, the fact that supplementation of Acetyl-CoA rescued MT acetylation in the Elp3cKO condition is suggestive of a metabolic defect leading

to the reduced production of Acetyl CoA. Accordingly, we performed new experiments to assess the expression of the enzymes that are responsible for the production of Acetyl-CoA in cells: *Acly* and *Acss2* (see also comments to the third reviewer) in cortical extracts from WT and *Elp3* cKO E14.5 embryos. The new figure 2 (panels f to h) now reports a 40% reduction of *Acly* and *Acss2* expression upon loss of *Elongator* activity. We decided to focus the proteomic analysis on the P3 fraction because we have previously shown that the *Atat1*-dependant MT acetylation is mostly driven by the vesicular fraction (1). Therefore, we performed brain fractionation and detected *Acly*, but not *Acss2*, in the vesicular fraction (P3) (see panels i to l of new Figure 2), which we confirmed is acting as the predominant driver of MT acetylation (panels O, P), at least in cortical cells. Immunolabeling and mass spectrometry confirmed the expression of *Acly* in vesicles, together with *Atat1* and *Elongator* subunits (new Figures 2m-n; S2A). Moreover, impairment of *Elongator* activity in *Elp1* KD mice resulted in the reduction of *Acly* expression in the brain vesicular fraction (Figure S2C). Since *Acss2* was also detected in the cytosolic fraction (S3), we suggest in the discussion its possible contribution to the production of Acetyl CoA in the vicinity of MT that could count for the residual level of MT acetylation seen upon axonal transport blockade in differentiated cortical neurons (see also(1)).

2/ A major finding of this study is the role of the *Elongator* complex in the regulation of *ACLY* protein levels, thereby identifying the cause of MT acetylation defects in *Elp3* KO cells and FD patient cells. While the authors argue that the identification of the mechanism for this is out of the scope of this study, they rule out an effect of *Elongator* on *ACLY* translation. To support this, they state that *ACLY* expression in mouse and fly *Elongator* models rescues the acetylation defects. This argument is however flawed, as defective translation could very well be rescued by strong overexpression of the *ACLY* transgene. What are the levels of human *ACLY* expression, as compared to endogenous *ACLY*? Ideally, an experiment disproving translation defects as a cause of the phenotype should be performed. This is particularly critical as protein translation is a major function of the *Elongator* complex. As stated in the previous comment, it would be important to estimate the fraction of proteins for which levels are defective.

We agree with the reviewer that our initial argument was not strongly supporting a lack of *Acly* mRNA translation defect in cortical neurons upon loss of *Elongator* activity. Moreover shedding some light on the mechanism leading to the reduced accumulation of *Acly* in *Elp3* cKO brains would strengthen the manuscript. Therefore, we combined two complementary approaches to test whether *Acly* mRNA translation was impaired upon lack of *Elongator* activity in the cortex. To address this question, we performed a polysomal fractionation from cortical extracts of P0 *Elp3* cKO and WT littermates to analyze the association of *Acly* mRNA with ribosomes and found no difference between genotypes (new Figures 5a-b). We confirmed these data by labeling cultured *Elp3* cKO and WT cortical PNs with a short pulse of puromycin, to metabolically label newly synthesized peptides. Cultured PNs were next immunolabelled with antibodies against *Acly* and puromycin (Figures 5c-d) and the subsequent puro-PLA assay used to visualize newly synthesized *Acly* peptides revealed comparable somatic and axonal puro-PLA *Acly* puncta for both genotypes (Figures 5e-f). Together, these data suggest that loss of *Elongator* activity does not interfere with the translation of *Acly* mRNA in cortical PNs. We also have novel data suggesting that the reduction of *Acly* expression reflects changes in protein stability upon *elongator* depletion

(new Figures 5g-h), likely resulting from the lack of interaction of Acly with Elongator subunits (new Figure S5).

3/ A precise description of how velocity quantifications have been performed is missing (in mat & met section). The reviewer understands that these graph representations may be classical, but the authors should clarify in the legend (or at least in the methods) what they mean by average velocity and moving velocity, and how this is measured. Also, for moving velocity representations, the authors should indicate to which direction correspond positive and negative velocities (I assume anterograde and retrograde but it should be indicated). Related to this, they should also explain why average velocities are represented in absolute values, while moving velocities are represented in positive and negative values. Finally, it is not clear how a pausing time can be expressed in percentage. Again, if this is standard, it should be explained in the legend. By percentage of pausing time, do they mean percentage of time spent in pause?

We thank the reviewer for this important comment and we are sorry for the lack of information provided in our manuscript regarding the analysis of axonal transport. We added the definition of Average and Moving Velocities together with the explanation of the anterograde vs retrograde orientation, the explanation of pausing time and the representation as negative, positive and absolute values in the Materials and Methods (page 32)

4/ While the quantifications are clear, the corresponding kymographs are often not very representative (Figures S1g, S3c, S3g...). The authors should replace some of them with examples that represent the quantifications more accurately. If the authors consider these as representative, they should highlight the events that are differently affected. Finally, and related to the previous comment, there are many large immobile structures. Are they all included in the quantification and how?

We have replaced most kymographs in the supplementary data, as suggest by the reviewer. Moreover, we did not include the large immobile structures in the analysis neither for the *in vivo* transport in larvae nor in cultures.

Specific comments

1/ The authors should quantify ACLY enrichment on the P3 fraction (Fig 2k).

This is indeed a great suggestion. The quantification of Acly enrichment, as well as Acss2, appears in new Figure 2k and the relative P3/S3 enrichment of these enzymes is shown in new Figure 2l.

2/ Figure S4h: The Proteostat signal appears brighter in the FD cells compared to control. Is the image not representative, or is there an effect on the size / number of aggregates per cell?

We agree with the reviewer, and we now show more representative pictures of aggregates that are also quantified (see new Figure S4i).

Minor comments

We are very sorry for the typos.

line 30: (MT)-based transport is an evolutionary conserved process

We modified accordingly

line 33: the mechanisms that control

We modified accordingly

line 34: remain

We modified accordingly

Line 249: "In extracts fibroblasts". Rather in fibroblast extracts

We modified accordingly

286: "is currently under investigation in the laboratory". The authors should avoid such statements.

We removed this statement

Figure 1a: is this experiment performed in a microfluidic device? The device is only described in 1b but the legend states 1a is a microfluidic experiment too.

Yes. PNs were also cultured in microfluidic device and this is now mentioned in the corresponding figure legend.

Figure 2i: Absorbance-

We corrected the figure.

Figure 2k: Although this is most likely to be due to a technical problem (during protein transfer for instance), the fact that there is almost no signal for Atax1 in the S1 fraction is incoherent. Ideally, another WB should be provided.

We have provided a new blot for the fraction enrichment of Atax1.

Figure 3a: The image for reduced acetylated Tubulin in Acl1 KD is not very convincing (especially when compared to 3f, which is clear). (Also, the scale bar is mispositioned.)

We thank the reviewer for the suggestion and we replaced the image with a more representative one.

Figure 4C: A very bright structure (both for Tub and ac-Tub) is shown in FD + Acly cells. What is it? If often observed, could it bias signal measurements?

This likely reflects a non-specific staining that we observed in some cells. It had no impact on the initial quantification of acetylated tubulin, which was performed in the cytoplasm and on another color channel. We have replaced this figure a more representative one.

Reviewer #2

Summary:

In this manuscript, the authors document a role for ATP-citrate lyase (ACLY) in the regulation of Atat1-dependent α -tubulin acetylation and axonal transport. Previous work identified a role for the Elongator complex in regulating microtubule acetylation, though mechanisms remained to be defined. Overall, the manuscript provides convincing evidence in support of their core claims that ACLY is present in vesicles along with Atat1, that its levels are reduced in the absence of Elongator, and that it regulates α -tubulin acetylation downstream of Elongator. Both murine and fly models are used. Very interestingly, relevance of this pathway in FD patient fibroblasts is shown.

We sincerely thank the reviewer for the constructive comments and suggestions aimed at improving the quality of our manuscript.

Major point:

1. Authors do a good job of documenting a key role for ACLY in the regulation of α -tubulin acetylation and axonal transport downstream of Elongator. However an important notion put forward here is that it is localized acetyl-CoA production by vesicular ACLY that is crucial for ATAT1-dependent tubulin acetylation. This is an important novel aspect of this paper, and in my view should be examined more critically.

The expertise of the reviewer on Acetyl-CoA metabolism is highly valued here. We initially focused our study on Acly because Acss2 was not detected in our vesiculome analysis. However, we agree that we should have analyzed the status of Acss2 expression to better understand the MT-acetylation defect seen upon loss of Elongator activity.

o ACLY and ACSS2 are the two major sources of cytosolic acetyl-CoA and ACSS2 can compensate for ACLY deficiency in at least some contexts. Why is ACLY critical for this function- is ACSS2 expressed? Is it present in vesicles?

To answer these key questions, we assessed the expression of both Acly and Acss2 in cortical extracts from E14.5 WT and Elp3 cKO embryos. The new figure 2 (panels f to h) reports a 40% reduction of Acly and Acss2 expression upon loss of Elongator activity. Thus, there was no compensation of Acly reduction by increased Acss2 expression. We next analyzed the fraction enrichment of these enzymes in WT cortical tissue and detected Acly, but not Acss2,

in the vesicular fraction (P3) (see panels i to l), which was confirmed as acting as the predominant driver of MT acetylation (panels O, P), at least in cortical cells. Immunolabeling and mass spectrometry confirmed the expression of Acly in vesicles, together with Atat1 and Elongator subunits (Figures 2m-n; S2A). Moreover, impairment of Elongator activity in Elp1 KD mice resulted in the reduction of Acly expression in the brain vesicular fraction (Figure S2C). Since Acss2 was also detected in the cytosolic fraction (S3), we discuss a possible contribution of Acss2 for the production of Acetyl CoA in the vicinity of MT that could account for the residual level of MT acetylation seen upon axonal transport blockade (see also(1)).

Does acetate supplementation rescue phenotypes in the absence of Elongator or ACLY? Does ACLY physically interact with ATAT1? Mews et al (Nature 2017) reported a key role for ACSS2 in regulating histone acetylation in neurons, and found it to be predominantly nuclear. Is it possible that the nuclear localization of ACSS2 precludes its involvement in a-tubulin acetylation regulation? Investigating these ideas would help to establish whether localized acetyl-CoA production is really a core part of this pathway.

Since Acss2 was also reduced in E14.5 Elp3 cKO cortical extracts, we did not test acetate supplementation. We performed co-IP to test the interaction between Acly and Atat1 and did not find any (not shown). However, we observed an interaction between ELP1/ELP3 and ACLY in transfected HEK293 cells (Figure S5). We were also surprised to detect no nuclear enrichment of Acss2 as previously reported by Mews et al. This discrepancy may indeed reflect the distinct origin (cortical neural cells versus Cath.-a-differentiated (CAD) cell line or hippocampal neurons) and/or state of differentiation of neurons analyzed (Acss2 has been shown to shuttle to the nucleus of CAD cells as they are maturing).

o This may not be feasible at this time, but the model could really be nailed down if ACLY could be localized away from vesicles, or if conditions that regulate ACLY vesicular localization could be identified, and loss of MT acetylation observed.

This is a great suggestion that would provide additional and more in depth mechanical understanding, but for which we would need a better molecular and structural knowledge of Acly that we are currently accumulating.

Minor:

1. I urge the authors to consider changing the title. "Fuels" implies a direct metabolic input into axonal transport, while the mechanism proposed is a post-translational modification. The language used in the running title seems more appropriate.

We agree and replaced the verb "fuels" by "promotes".

2. Please clarify in figures or legends the conditions being compared in each statistical analysis.

Thank for the suggestion, we clarified this in figure legends, by adding the following statement where relevant: "... the post hoc multiple comparisons, to analyze statistical difference of each condition compared with control... «

3. Fig. 1K: Elp3 KD;ATAT1 KD appear to have additive effects. Please also include statistics comparing this to Elp3 KD alone.

We performed the request a statistical analysis on the figure for comparing these two conditions and could not detect any ($p=0.0637$).

4. Perhaps I missed it, but please blot for ATAT1 comparing WT to ATAT1 KO. There is sometimes a double band and sometimes not in the blots shown for ATAT1, and it would be helpful to clarify whether one or both of the bands are ATAT1.

In mice brain cortex two Atat1 isoforms can be detected: #3 and #4, corresponding to 30KD and 37 KD, respectively. These isoforms have previously been discussed in a former paper from our laboratory (Even et al., 2019. Science advances (1)). We added some clarification in the figure legends together with the corresponding reference.

5. Fig 3i: absorbance is misspelled in y axis

Thank you for noticing the typo, we corrected it in the corresponding figure.

Reviewer #3

The manuscript by Even et al. claims to identify ATP-citrate lyase (Acy) as a rate-limiting enzyme that modulates activity of Atat1, which acetylates alpha-tubulin. They show that neurons lacking Elongator complex have decreased Acy levels which contributes to deficits in tubulin acetylation and MT-dependent transport in projection neurons. They further report similar deficits in MT-dependent transport in fibroblasts from Familial Dysautonomia (FD) patients carrying mutations in Elongator complex. While this previously unknown role of Acy in the function of Elongator complex acetylating tubulin is an important finding and could be of general interest to the cytoskeleton field, I have several major concerns for the manuscript as it stands now, outlined below:

We thank the reviewer for the encouraging comments and we have done several new experiments to answer the questions raised below.

1. The authors perform bulk of their most important measurement of the ratio of acetylated α -tubulin to total tubulin via immunofluorescence (IF) images. Since they do have a good antibody for acetylated α -tubulin (as shown in supplementary Figure S1) to detect in western blots, and given that extracting protein measurements using IF images can be tricky and susceptible to huge variations across samples owing to different staining levels, it is important that the authors complement the IF data with quantitation using western blots.

We agree that it is not always easy to quantify immunolabeling intensity. We followed the suggestion of the reviewer and perform complementary western blotting analyses. We are now reporting a significant reduction of acetylation of alpha tubulin (normalized on total

alpha tubulin) in cortical extracts from E14.5 Elp3 cKO mouse embryos in the new Figure S1a, to complement the immunolabeling analysis in Figure 1a. A similar experiment was done with micro-dissected Elp3KD fly heads (recombination in the brain of UAS:Elp3 X Elav:Gal4 flies), for which we report a matching reduction of alpha tubulin acetylation (normalized on total alpha tubulin) when comparing the immunoblotting (Figure S1r) and immunolabeling (Figure 1f) analysis. Finally, we also complemented the analysis for human primary fibroblasts matching the immunolabeling of S4c to S4b.

2. For some of the axonal transport data shown in main Figures, the authors never provide any raw data in the form of kymographs, as in Figures 1c,d. It is imperative that the authors provide raw data for all of their quantification in the main figures.

We agree and the kymographs for figure 1c,d are shown in the new figure S1o.

Furthermore, in the kymographs shown in the supplemental figures – kymographs for some of the rescue experiments do not really reflect the phenotype being reverted back to WT. For example, HDAC6 KD rescue kymographs in Figure S1q Elp3 KD, Acly rescue kymograph in Figure S3c Acly rescue kymographs in Figure S3g None of these kymographs represent the supporting quantitation and the claims the authors make in the text. Are these just a one-off? If yes, can the authors pick more representative images. Or are there concerns about the phenotypes not being robust enough?

We agree and we replaced these specific kymographs with more representative ones.

3. I have several comments on Figure 2.

(i) I am surprised with the result in Figure 2d where the authors report tubulin acetylation level of Elp3 cKO cortical extracts comparable to WT extracts. If Elongator complex is required for tubulin acetylation, how do you explain this result?

The explanation is that we added Acetyl-CoA in this experiment, which is indeed reduced upon loss of Elongator activity and is responsible for the reduction of MT acetylation (Atat1 level is not changing) (see also new Figure 2u).

(ii) Does lack of Elp3 affect tubulin de-acetylation as well as shown in Figure 2e? The authors should perform this experiment with Atat1 KO as well, as done in Figure 2d

The lack of Elp3 does not impair alpha tubulin de-acetylation (see new Figure 2d). We respectfully disagree with the reviewer on the interest of doing a comparable experiment with Atat1KO extracts because Atat1 is the only driver of MT acetylation without which there is no acetyl groups to be removed by HDAC6 from tubulin dimers.

(iii) In addition to the data in Figures 2m, n the authors should also show Acly levels in the whole extract of WT vs Elp3 cKO to determine if the overall levels of Acly are also affected?

This is a great suggestion and we performed new western blots on total cortical extract from E14.5 WT or Elp3 cKO embryos showing that Acly expression is reduced by 37% upon loss of Elp3 expression (new Figures 2f, 2h).

4. How do the authors explain rescue of axonal deficits in Elp3 KO neurons with HDAC6 KD? This is an interesting result and is worth some discussion in the paper – if the authors think axonal transport is determined by a fine balance of acetylated vs de-acetylated tubulin? If this model is in fact true, can HDAC6 KD revert the axonal deficits they report in Acly KD neurons shown in Figure 3?

Again, we thank the reviewer for suggesting this additional set of key experiments. The reviewer guessed well as our new data show a full rescue of synaptic vesicles axonal transport parameters in 3rd instar larvae as well as motor behaviors (in both larvae and adult flies) upon depletion of Acly, by simultaneously knocking down Hdac6 (Figure 3e-h). We can therefore conclude that the completion of those biological processes requires a proper level of MT acetylation.

5. The authors constantly keep switching between in vivo imaging of Drosophila neurons and neuronal cultures from various mouse lines. While the results being consistent across species is encouraging, the differences in experiments performed across these two model systems sometimes makes the results harder to understand/interpret. For example, in Figure 1 the authors start out with neurons from Elp3 cKO mice and then switch to experiments with neurons from Atat1 KO mice harboring RNAi for Elp3. They perform these experiments in the fly model as well and rescue the phenotype with KD of Hdac6, which is not done with neurons from Elp3 cKO mice. The authors finish up the data set with Elp3 cKO mice again in supplementary figure S1. This seems to be the case across multiple figures throughout the paper. Can the authors consolidate the data across figures that helps with data interpretation and will also make it easier for the reader.

Our decision to use distinct models was to provide a strong evidence for a conserved pathway for MT acetylation across species and evolution, but has the disadvantage to require a large amount of work to repeat all experiments. In order to avoid extensive animal breeding and to limit animal sacrifice (as requested by our local ethical committee who approved the experimental protocol of this study) we have not performed the following breeding : Elp3^{lox/lox} X FoxG1:cre X Atat1 KO to obtain triple transgenics in order to replicate the data obtained by expression of sh-Elp3 in Atat1 KO PNs (new Figures 1c-e). The efficiency of the sh-Elp3 construct has been validated (new Figures S1i-j).

Regarding the missing experimental piece showing a possible Hdac6KD rescue of the transport defects in Elp3 cKO mouse PNs (shown in new Figures S1b-h) as shown in flies. As we do not have access to the Hdac6 KO mouse line, we tested supplementation of Elp3 cKO PNs with tubastatin (to block Hdac6 activity) but we were unfortunately not able to maintain these neurons in good shape to record axonal transport.

However, in order to consolidate our data set, we now provide transport data in Elp3 cKO PNs cultured neurons treated with the Acly inhibitor hydroxycitric acid (HCA) (new Figures 3a-d) that nicely recapitulate the axonal transport phenotype observed in AclyKD flies (new Figure s3e-h).

1. Even A, Morelli G, Broix L, Scaramuzzino C, Turchetto S, Gladwyn-Ng I, et al. ATAT1-enriched vesicles promote microtubule acetylation via axonal transport. *Sci Adv.* 2019;5(12):eaax2705.

REVIEWERS' COMMENTS

Reviewer #1 (Remarks to the Author):

The authors have largely addressed my concerns and I am now happy to recommend the article for publication.

Reviewer #2 (Remarks to the Author):

The authors have addressed my concerns and I recommend publication.

Reviewer #3 (Remarks to the Author):

The authors have addressed all my concerns and the organization of the figures and the text are much clearer in this version. I recommend publication!